# Targeting A-kinase anchoring protein 12 phosphorylation in hepatic stellate cells regulates liver injury and fibrosis in mouse models

Komal Ramani[1,2]*, Nirmala Mavila[1,2]*, Aushinie Abeynayake[1],
Maria Lauda Tomasi[1,2]*, Jiaohong Wang[1], Michitaka Matsuda[1], Eki Seki[1,2]

[1]Karsh Division of Gastroenterology and Hepatology, Cedars-Sinai Medical Center, Los Angeles, United States; [2]Applied Cell Biology Division, Department of Biomedical Sciences, Cedars-Sinai Medical Center, Los Angeles, United States

**Abstract** Trans-differentiation of hepatic stellate cells (HSCs) to activated state potentiates liver fibrosis through release of extracellular matrix (ECM) components, distorting the liver architecture. Since limited antifibrotics are available, pharmacological intervention targeting activated HSCs may be considered for therapy. A-kinase anchoring protein 12 (AKAP12) is a scaffolding protein that directs protein kinases A/C (PKA/PKC) and cyclins to specific locations spatiotemporally controlling their biological effects. It has been shown that AKAP12's scaffolding functions are altered by phosphorylation. In previously published work, observed an association between AKAP12 phosphorylation and HSC activation. In this work, we demonstrate that AKAP12's scaffolding activity toward the endoplasmic reticulum (ER)-resident collagen chaperone, heat-shock protein 47 (HSP47) is strongly inhibited by AKAP12's site-specific phosphorylation in activated HSCs. CRISPR-directed gene editing of AKAP12's phospho-sites restores its scaffolding toward HSP47, inhibiting HSP47's collagen maturation functions, and HSC activation. AKAP12 phospho-editing dramatically inhibits fibrosis, ER stress response, HSC inflammatory signaling, and liver injury in mice. Our overall findings suggest a pro-fibrogenic role of AKAP12 phosphorylation that may be targeted for therapeutic intervention in liver fibrosis.

*For correspondence:
komal.ramani@cshs.org (KR);
nirmala.mavila@cshs.org (NM);
marialauda.tomasi@cshs.org
(MLT)

**Competing interest:** The authors declare that no competing interests exist.

## Editor's evaluation

Liver fibrosis is a complication of diverse liver disorders, including fatty liver disease, and in this important work, Ramani et al. provide solid evidence that AKAP12 enhances collagen production in a CCL4-induced mouse model of liver fibrosis, and they demonstrate that in hepatic stellate cells (HSCs) AKAP12 is phosphorylated by PKCalpha, which, in turn, leads to increased scaffolding activity towards HSP47, a chaperone of collagen located in the endoplasmic reticulum (ER). AKAP12 activation also resulted in increased ER stress and the generation of inflammatory mediators, and targeting AKAP12 phosphorylation sites in HSCs resulted in suppression of collagen synthesis and ER stress and fibrotic response in the mice. The important findings of the study include the identification of a novel disease mechanism and an attractive drug target for liver fibrosis. The finding should, however, be cautiously interpreted as the role of AKAP12 in liver fibrosis has to be explored in other models of liver fibrosis.

## Introduction

Hepatic stellate cells (HSCs) constitute approximately 5–8% of the normal liver and are major sites for vitamin A storage in the body (*Friedman, 2008*). During chronic liver injury, HSCs acquire a pro-fibrogenic phenotype or activated state that is critical in the liver's response to injury (*Hernández-Gea*

*et al., 2013*). HSC activation causes increased production of extracellular matrix (ECM) components such as collagens and α-smooth muscle actin (α-SMA). Persistent injury leads to fibrosis due to abnormal accumulation of ECM (*Li et al., 2008*). HSC pathways that cause fibrogenic responses in the liver can be targeted for therapeutic intervention.

Collagen maturation and secretion are facilitated by the endoplasmic reticulum (ER)-resident chaperone, heat shock protein 47 (HSP47) along with other ER foldases such as BIP/GRP78 (*Kawasaki et al., 2015*; *Sepulveda et al., 2018*). Under normal physiological conditions, HSP47 is expressed at low levels in the liver (*Brown et al., 2005*) and other organs such as lung, heart, and kidney (*Khalil et al., 2019*). Fibrogenic stimulation by carbon tetrachloride ($CCl_4$) or bile duct ligation (BDL) in mice and human liver fibrosis is associated with induction of HSP47 expression (*Brown et al., 2005*; *van de Bovenkamp et al., 2005*; *Xia et al., 2006*). The induction in HSP47 correlates with increased collagen secretion from activated HSCs during liver fibrosis. Therefore, silencing HSP47 to inhibit collagen production is an appealing option for reversing fibrosis (*Thompson et al., 2011*). However, because HSP47 also plays a chaperoning function in the healthy liver and other organs, the collateral effects of its therapeutic silencing should be investigated (*Thompson et al., 2011*). Apart from its function as a collagen chaperone, a recent interactome study identified HSP47 as a binding partner for an unfolded protein response (UPR) sensor protein, inositol-requiring enzyme 1 alpha (IRE1α) (*Sepulveda et al., 2018*). HSP47 activates IRE1α oligomerization and phosphorylation by displacing its regulator, BIP, thereby triggering the UPR response during ER stress (*Sepulveda et al., 2018*). Whether triggering of UPR signaling by HSP47-IRE1α interaction and BIP displacement in HSCs may enhance the folding of pro-fibrogenic proteins such as collagen is unclear so far. But it is generally accepted that HSCs exhibit ER stress and UPR signaling in response to liver injury stimuli (*Maiers and Malhi, 2019*).

A-kinase anchoring protein 12 (AKAP12) is a ubiquitously expressed member of the AKAP family that exhibits scaffolding activity toward signaling molecules including protein kinases (PKA and PKC), β2-adrenergic receptor, cyclins-(cyclin-D1, CCND1) (*Gelman, 2002*), and polo-like kinase 1 (PLK1) (*Canton et al., 2012*). By virtue of its scaffolding function, AKAP12 spatiotemporally controls cellular signaling by guiding its binding partners to their physiological substrates or specific functional locations (*Tröger et al., 2012*). These activities regulate growth, cytoskeletal remodeling, and adrenergic signal transduction (*Gelman, 2002*). AKAP12-mediated scaffolding of PKC attenuates PKC activation and suppress oncogenic proliferation, invasiveness, chemotaxis, and senescence (*Akakura et al., 2010*; *Akakura and Gelman, 2012*). PKCα, δ, and ε isoforms interact with AKAP12, however only PKCα and δ activity is induced in the absence of AKAP12 (*Guo et al., 2011*; *Su et al., 2010*). AKAP12 sequestration of CCND1 in the cytoplasm prevents its nuclear translocation and cell cycle progression (*Lin et al., 2000*). AKAP12 sequestering of CCND1 and inhibition of CCND1 activity have been reported in parietal glomerular epithelial cells and in fibrosarcoma (*Yoon et al., 2007*; *Burnworth et al., 2012*).

It has been demonstrated that the scaffolding ability of AKAP12 is altered by its phosphorylation (*Guo et al., 2011*; *Lin et al., 2000*; *Xia and Gelman, 2002*). Prephosphorylation of AKAP12 by PKC suppresses its interaction with PKC itself and increases PKC activity (*Gelman, 2010*). Phosphorylation of AKAP12 at a PKC phosphorylation site (S507/515) prevents the sequestration of CCND1 by AKAP12 leading to its nuclear translocation, allowing cell cycle progression (*Lin et al., 2000*; *Burnworth et al., 2012*). AKAP12 phosphorylation by cyclin-dependent kinase 1 (CDK1) at a threonine residue (T766) enhances the recruitment of the polo-like kinase (PLK1) in human glioblastomas to ensure efficient mitotic progression (*Canton et al., 2012*). Even though phosphorylation is known to regulate AKAP12's scaffolding activities, the functional impact of its phospho-modifications on liver disease has not been evaluated. We previously demonstrated that HSC activation during liver injury was associated with an induction in phospho-AKAP12 (*Ramani et al., 2018*). In this work, we demonstrate that specific AKAP12 phosphorylation events in HSCs regulate its scaffolding activity toward the collagen chaperone, HSP47. HSC-specific CRISPR-editing of AKAP12's phospho-sites preserves the AKAP12-HSP47 scaffold, reduces HSP47's collagen-chaperoning activity, dramatically lowering overall collagen content and liver injury during carbon-tetrachloride ($CCl_4$)-induced liver fibrosis. AKAP12 phospho-modulation directed toward HSCs regulates HSP47-IRE1α interaction, thereby controlling UPR signaling in HSCs. Furthermore, AKAP12 phospho-site modulation in HSCs suppresses overall ER stress in the fibrotic liver. Our data support a previously unidentified function of AKAP12 and its phospho-modification in regulating the outcome of liver fibrosis in animal models.

## Results

### Expression, phosphorylation, and scaffolding activity of AKAP12 is altered in CCl₄-treated mouse liver and human liver fibrosis

The expression of AKAP12 protein was decreased in livers of $CCl_4$-treated mice by 13% compared to oil controls (*Figure 1A*, left panel) without a change in *Akap12* mRNA (*Figure 1A*, right panel). $CCl_4$ treatment induced the expression of HSC activation marker, *Acta2* by 1.5-fold and its corresponding protein, α-SMA, by 6.4-fold compared to control (*Figure 1A*, *Figure 1—source data 1*). As evidenced by proximity ligation assay (PLA), the phosphorylation of AKAP12 was induced in desmin-positive HSCs of $CCl_4$ livers by fivefold compared to control (*Figure 1B*, *Figure 1—source data 2*). AKAP12 staining judged by ImageJ quantification (see Materials and methods) was decreased in $CCl_4$-treated liver by 16% compared to control (*Figure 1B*, *Figure 1—source data 2*) consistent with the western blot result (*Figure 1A*). The interaction of AKAP12 with HSP47 was inhibited by 54% despite a 3.9-fold increase in overall HSP47 levels in $CCl_4$ livers compared to control (*Figure 1C*, *Figure 1—source data 3*). A human liver fibrosis tissue array containing 16 liver fibrosis tissues and 11 normal tissues was stained with PLA probes for AKAP12 and HSP47 to detect their interaction. The interaction between AKAP12 and HSP47 was inhibited by 68% in human liver fibrosis tissue compared to normal (*Figure 1D*, *Figure 1—figure supplement 1*). This was associated with a 20% decrease in total AKAP12 staining and a 3.8-fold increase in HSP47 staining in liver fibrosis compared to normal (*Figure 1D*). Post hoc analysis of *Figure 1A–D* is provided in *Figure 1—source data 4*.

### CRISPR-directed editing of AKAP12's activation-responsive phospho-sites enhances AKAP12's scaffolding activity and inhibits HSC activation

The phospho-peptide map of AKAP12 protein from Day 7 culture-activated human or mouse HSCs was compared to that of Day 0 quiescent HSCs or normal hepatocytes (*Table 1*). A peptide region containing 5S/T phospho-sites exhibited increased phosphorylation in Day 7 activated HSCs but not in Day 0 HSCs or hepatocytes (*Table 1—source data 1*). These activation-responsive phospho-sites were conserved in mouse and human (*Table 1*). The interaction between AKAP12 and HSP47 was reduced by 40% after 3 days and by 86% after 6 days of HSC culture-activation compared to Day 0 (*Figure 2A*, *Figure 2—source data 1*). This was associated with a corresponding induction in the levels of α-SMA up to 5.8-fold by Day 6 compared to Day 0 (*Figure 2A*, *Figure 2—source data 1*). Day 5 activated human HSCs were transfected with CRISPR small guide RNA (sgRNA) and donor RNA (see Key resource table) to delete the five AKAP12 phosphorylation sites by homology-directed repair (HDR) as described under Materials and methods. Genomic DNA PCR from CRISPR edited (HDR) cells using deletion-specific primers (see Key resource table) resulted in a 261-bp amplicon that was not amplified in wild-type (WT) cells or cells treated with SaCas9 (*Staphylococcus aureus* CRISPR-associated protein) alone (*Figure 2B*, original gel shows four experiments). The interaction between AKAP12 and HSP47 in CRISPR-edited HSCs (HDR) was induced by 2.5-fold compared to WT cells (*Figure 2C*, original blot developed with anti-mouse IgG is shown in *Figure 2—source data 2*). This was associated with a 40% decrease in α-SMA levels, demonstrating that AKAP12 phospho-site editing inhibited HSC activation (*Figure 2C*). The overall level of HSP47 decreased by 25% whereas the level of AKAP12 protein remained unchanged after HDR (*Figure 2C*). Deletion of phospho-sites in mouse HSCs resulted in a 422-bp deletion-specific amplicon (*Figure 2—figure supplement 1*, *Figure 2—figure supplement 1—source data 1*). Like human HSCs, mouse HSCs also exhibited increased AKAP12-HSP47 interaction after AKAP12 phospho-site editing (*Figure 2—figure supplement 1*, *Figure 2—figure supplement 1—source data 2*). Reversal of HSC activation by AKAP12 editing was determined by examining vitamin A auto fluorescence (*Senoo et al., 2010*). Cultured human HSCs at Day 0 exhibited strong vitamin A autofluorescence that was reduced in Day 5 activated HSCs (*Figure 2D*, three independent experiments are shown). AKAP12 editing in Day 5 HSCs restored the loss of vitamin A fluorescence compared to Day 5 HSCs or Day 5 HSCs+SaCas9 alone (*Figure 2D*). HSP47 is an ER-resident chaperone (*Kawasaki et al., 2015*). A weak PLA signal of AKAP12-HSP47 interaction co-localized with the ER marker, calreticulin in activated (WT) HSCs (*Figure 2E*, left panel). However, upon CRISPR-editing (HDR), a strong AKAP12-HSP47 PLA signal co-localized with calreticulin in the ER (*Figure 2E*, left panel, *Figure 2—source data 3*). We examined whether HSP47's

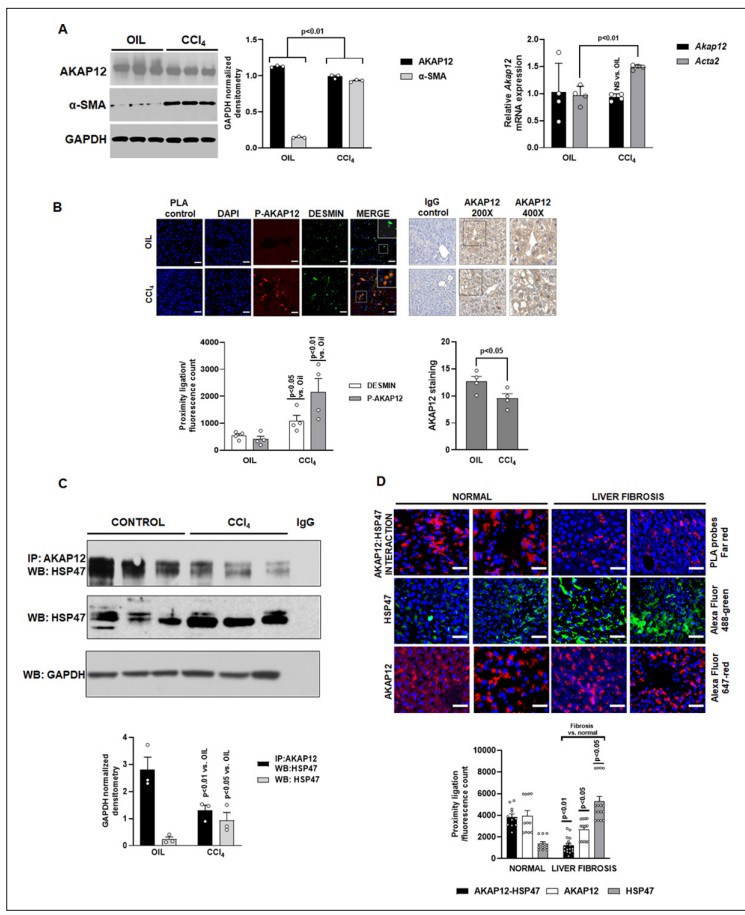

**Figure 1.** Expression, phosphorylation, and scaffolding activity of AKAP12 is altered in CCl$_4$-treated mouse liver and human liver fibrosis. Mice were administered CCl$_4$ or mineral oil (control) as in methods. (**A**) Total protein (left panel) was immunoblotted with AKAP12, α-SMA, or GAPDH (control) antibody and blots were quantified by ImageJ densitometry. Data represented by GAPDH normalized densitometry is mean ± SE from three experimental groups. Source data are presented in *Figure 1—source data 1*. P values are calculated in *Figure 1—source data 4*. Total RNA (right panel) from mouse liver was subjected to real-time RT-PCR to evaluate the expression of *Akap12, Acta2*, or *Gapdh* (normalizing control) mRNA (mean ± S.E from four experimental groups). P values are calculated in *Figure 1—source data 4*. (**B**) Sections of control or CCl$_4$ livers stained with the HSC marker, desmin was overlayed with antibodies to detect ligation of AKAP12 with the phospho-serine antibody by PLA as in methods. 200× magnification, scale bar=50 μm. Total AKAP12 expression was detected by HRP/DAB staining as in Materials and methods. Images were quantified using ImageJ and represented as the proximity ligation/fluorescence/HRP count. Mean ± SE from four experimental groups. Source data are presented in *Figure 1—source data 2*. P values are calculated in *Figure 1—source data 4*. (**C**) Control or CCl$_4$ liver protein was immunoprecipitated with AKAP12 antibody and probed for HSP47 by western blotting. Normal mouse IgG was a negative control. Data represented by GAPDH normalized densitometry are mean ± SE from three experimental groups. Source data are presented in *Figure 1—source data 3*. P values are calculated in *Figure 1—source data 4*. (**D**) Human tissue arrays were stained with AKAP12 and HSP47 far red PLA probes as in methods. AlexaFluor antibodies (see Key resource table) were used to detect expression of AKAP12 or HSP47 in these arrays. A representative area is shown at 400× magnification, scale bar=100 μm. Each tissue within the array was quantified by densitometry using ImageJ and represented as the proximity ligation/fluorescence count (*Figure 1—figure supplement 1*). Mean ± SE, from 11 normal livers and 16 liver fibrosis tissues. P values are calculated in *Figure 1—source data 4*.

The online version of this article includes the following source data and figure supplement(s) for figure 1:

**Source data 1.** Raw blots for *Figure 1A*.

**Source data 2.** Individual images for *Figure 1B*.

**Source data 3.** Raw blots for *Figure 1C*.

**Source data 4.** Post hoc analysis for *Figure 1*.

*Figure 1 continued on next page*

*Figure 1 continued*

**Figure supplement 1.** Complete human tissue arrays of 11 normal livers and 16 liver fibrosis tissues stained with PLA probes to detect AKAP12-HSP47 interaction and Alexa fluor probes to detect HSP47 (green) or AKAP12 (red) as described under Materials and methods.

collagen-chaperoning activity was regulated by AKAP12 phospho-site editing. Our results show that the collagen-HSP47 PLA signal strongly co-localized in the ER of activated HSCs (WT) (*Figure 2E*, right panel). CRISPR-editing of AKAP12 (HDR) reduced the collagen-HSP47 interaction significantly by 65% compared to WT cells (*Figure 2E*, right panel, *Figure 2—source data 3*). Post hoc analysis of *Figure 2A, C and E* is provided in *Figure 2—source data 4*.

## PKCα phosphorylates AKAP12 and inhibits its interaction with HSP47

Kinase-prediction software was used to predict that out of the five AKAP12 activation-responsive phospho-sites, two serine residues (S687/S688) were strongly predicted substrates of PKCα kinase with a consensus of [S/T]-X-R/K whereas one threonine (T675) could not be assigned a kinase (*Supplementary file 1*). S676/S678 sites were also PKCα sites but shared consensus sites with calmodulin kinase (CAMK). The overall confidence of prediction for the S676/S678 sites was less than that of S687/S688 sites. In vitro kinase assay followed by phostag gel analysis revealed that phosphorylation of AKAP12 was significantly enhanced in the presence of active PKCα enzyme compared to kinase negative controls (*Figure 3A*). Mutations of AKAP12 S676/S678 to alanine modestly reduced the phosphorylation of biotinylated recombinant AKAP12 whereas S687A/S688A mutation dramatically suppressed the phostag shift of AKAP12 (*Figure 3A*). The mutation seemed to completely inhibit the phospho-band. Since other phosphorylation events could also cause the shift, we repeated the experiment to see whether this complete suppression was reproducible. In an additional experiment (*Figure 3—source data 1*), we observed that the S687/S688A mutation suppressed but did not always wipe out the phospho-shift. Also, in some experiments, we observed the -kinase control had a faint phospho-signal. The recombinant protein produced by RRLs in an in vitro translated system may have baseline phosphorylation as reported in the manufacturer's protocol (TNT Coupled Transcription/Translation system, Promega). Direct binding was observed between biotinylated AKAP12 and HSP47 in a recombinant system in the absence of active PKCα (*Figure 3B*, *Figure 3—source data 2*). Presence of PKCα inhibited the interaction between AKAP12 and HSP47 (*Figure 3B*). To evaluate whether

**Table 1.** Phospho-peptide mapping of human HSCs, mouse HSCs, and mouse hepatocytes.

| Cell type | Observed precursor mass | Neutral loss of phosphate mass | Phospho-peptide sequence | Peptide modification |
|---|---|---|---|---|
| Day 0 human HSC | 1988.7812 | 1890.0297 | KRKVDTSVSWEALICVGS**S**KK | Phospho (ST)[16] |
| Day 7 human HSC | 2148.9758 | 1854.9 | KRKVD**T**SVSWEALICVG**SS**K | Phospho (ST)[16,17], |
| Day 7 human HSC | 2148.9932 | 1855.3141, 1854.9 | KRKVD**T**S**V**SWEALICVGS**S**KK | Phospho (ST)[4,6,16] |
| Day 0 mouse HSC | ND | ND | KRKVDTSVSWEALICVGSSKK | ND |
| Day 7 mouse HSC | 1998.13 | 1801.7952 | KRKVD**T**SV**S**WEALICVGSSK | Phospho (ST)[3,6] |
| Day 7 mouse HSC | 2054.22 | 1857.9027 | KRKVDTSVSWEALICVG**SS**KK | Phospho (ST)[14,15] |
| Mouse hepatocytes | ND | ND | KRKVDTSVSWEALICVGSSK | ND |

S=Serine, T=Threonine, ND=not detected.

The online version of this article includes the following source data for table 1:

**Source data 1.** Phospho-peptide map for *Table 1*.

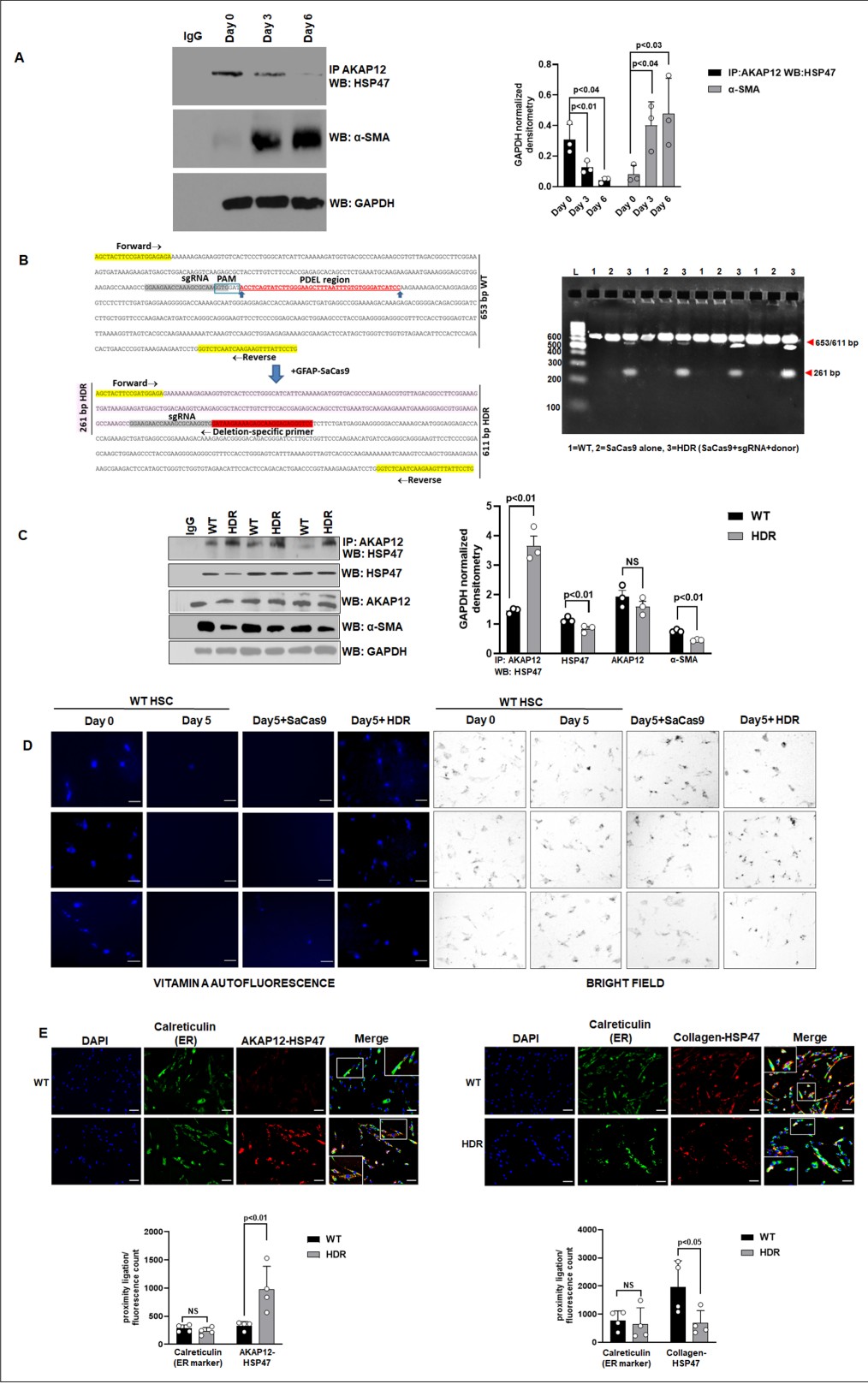

**Figure 2.** CRISPR-directed editing of AKAP12's activation-responsive phospho-sites enhances AKAP12's scaffolding activity and inhibits HSC activation. (**A**) Cell extracts from human HSCs cultured for 0, 3, or 6 days (see Materials and methods) were processed for co-immunoprecipitation of AKAP12 and HSP47 or for α-SMA western blotting. Data represented as GAPDH normalized densitometry is mean ± SE from three experiments. Source

*Figure 2 continued*

data are presented in *Figure 2—source data 1*. P values are calculated in *Figure 2—source data 4*. (**B**) Activated human HSCs were transfected with CRISPR reagents and GFAP-SaCas9 vector to cause CRISPR-directed HDR as in Materials and methods. Untransfected (WT) or cells with SaCas9 alone were used as controls. CRISPR editing at the *AKAP12* locus (left panel) was confirmed by performing PCR (right panel) using primers that specifically detected the edited region as listed in Key resource table. Four independent experiments are shown. (**C**) CRISPR-edited (HDR) or WT cells as in (**B**) above were assessed for AKAP12-HSP47 co-immunoprecipitation, HSP47, AKAP12, and α-SMA (HSC activation marker) western blotting. Data represented as GAPDH normalized densitometry is mean ± SE from three experiments. Source data are presented in *Figure 2—source data 2*. P values are calculated in *Figure 2—source data 4*. (**D**) Day 0 attached HSCs were culture activated till Day 3 and then transfected with CRISPR vectors till Day 5. The autofluorescence of vitamin A as a marker of HSC quiescence was visualized by fluorescence microscopy and compared to brightfield images of cells as in Materials and methods. Three independent experiments are shown. Scale bar=80 µm. Source data is presented in *Figure 2—source data 5*. (**E**) AKAP12-HSP47 interaction (left panel) and HSP47-collagen interaction (right panel) in the ER was compared between WT and HDR cells by PLA staining and co-staining with the ER marker, calreticulin as in methods. Magnification at 200×, scale bar=60 µm. Data represented as proximity ligation/fluorescence count are mean ± SE from four experiments. Source data are presented in *Figure 2—source data 3*. P values are calculated in *Figure 2—source data 4*. ER, endoplasmic reticulum; HDR, homology-directed repair; HSC, hepatic stellate cell; PLA, proximity ligation assay; WT, wild-type.

The online version of this article includes the following source data and figure supplement(s) for figure 2:

**Source data 1.** Source blots for *Figure 2A*.

**Source data 2.** Source blots for *Figure 2C*.

**Source data 3.** Source data for *Figure 2E*.

**Source data 4.** Post-hoc analysis for *Figure 2*.

**Source data 5.** Source data for *Figure 2D*.

**Figure supplement 1.** CRISPR-directed editing of AKAP12's activation-responsive phospho-sites enhances AKAP12's HSP47 scaffolding activity in mouse HSCs.

**Figure supplement 1—source data 1.** Source blots (*Figure 2—figure supplement 1—source data 2*).

**Figure supplement 1—source data 2.** Original gel image.

phosphorylation of AKAP12 by PKCα in HSCs would regulate AKAP12's scaffolding activity, cells were treated with *Prkca* siRNAs (A or B). Silencing *Prkca* by 74% with siRNA-A increased AKAP12-HSP47 interaction by threefold whereas a 90% knockdown caused by siRNA-B enhanced AKAP12-HSP47 interaction by eightfold compared to negative control siRNA (*Figure 3C*, *Figure 3—source data 3*). HSP47 levels remain unchanged by siRNA treatments (*Figure 3C*). Post hoc analysis of *Figure 3C* is provided in *Figure 3—source data 4*.

## In vivo gene editing of the *Akap12* region corresponding to its activation-responsive phospho-sites in HSCs of mouse liver

The *Akap12* exon 3 contains sequences corresponding to the activation-responsive phospho-sites of AKAP12 protein. To perform gene editing of this region specifically in HSCs of mouse liver, two different CRISPR HDR approaches were used (*Figure 4A*). A PDEL donor was used to delete the AKAP12 phospho-sites whereas each S or T phospho-site was mutated to A using a PMUT donor. Two unique sgRNAs specific for the region around the phospho-sites along with the donor (*Figure 4A*, Key resource table) were cloned into AAV vectors (*Figure 4B*, left panel). To perform CRISPR editing in HSCs of mouse liver, the SaCas9 enzyme was cloned into AAV vector under control of two different HSC-specific promoters (Glial fibrillary acidic protein, GFAP or Lecithin retinol acyltransferase, LRAT) (*Puche et al., 2013*; *Lee et al., 2020*). AAV vectors were injected into mice during oil or $CCl_4$ administration according to the plan in *Figure 4B*, right panel. To evaluate the HSC specificity of GFAP-SaCas9 mediated CRISPR (CR) editing compared to that of an empty vector (EV) control (see Materials and methods), genomic DNA of HSCs or hepatocytes isolated from livers of oil+EV, oil+CR, $CCl_4$+EV, and $CCl_4$+CR groups was subjected to multiplex PCR with PDEL forward and reverse primers and a PDEL deletion-specific primer (see Key resource table). Oil+EV or $CCl_4$+EV HSCs or hepatocytes gave a 298-bp amplicon in this multiplex PCR. Oil+CR or $CCl_4$+CR groups resulted in 298 bp WT and 256 and 154 bp mutated amplicons due to complementarity with the deletion-specific primer (*Figure 4C*,

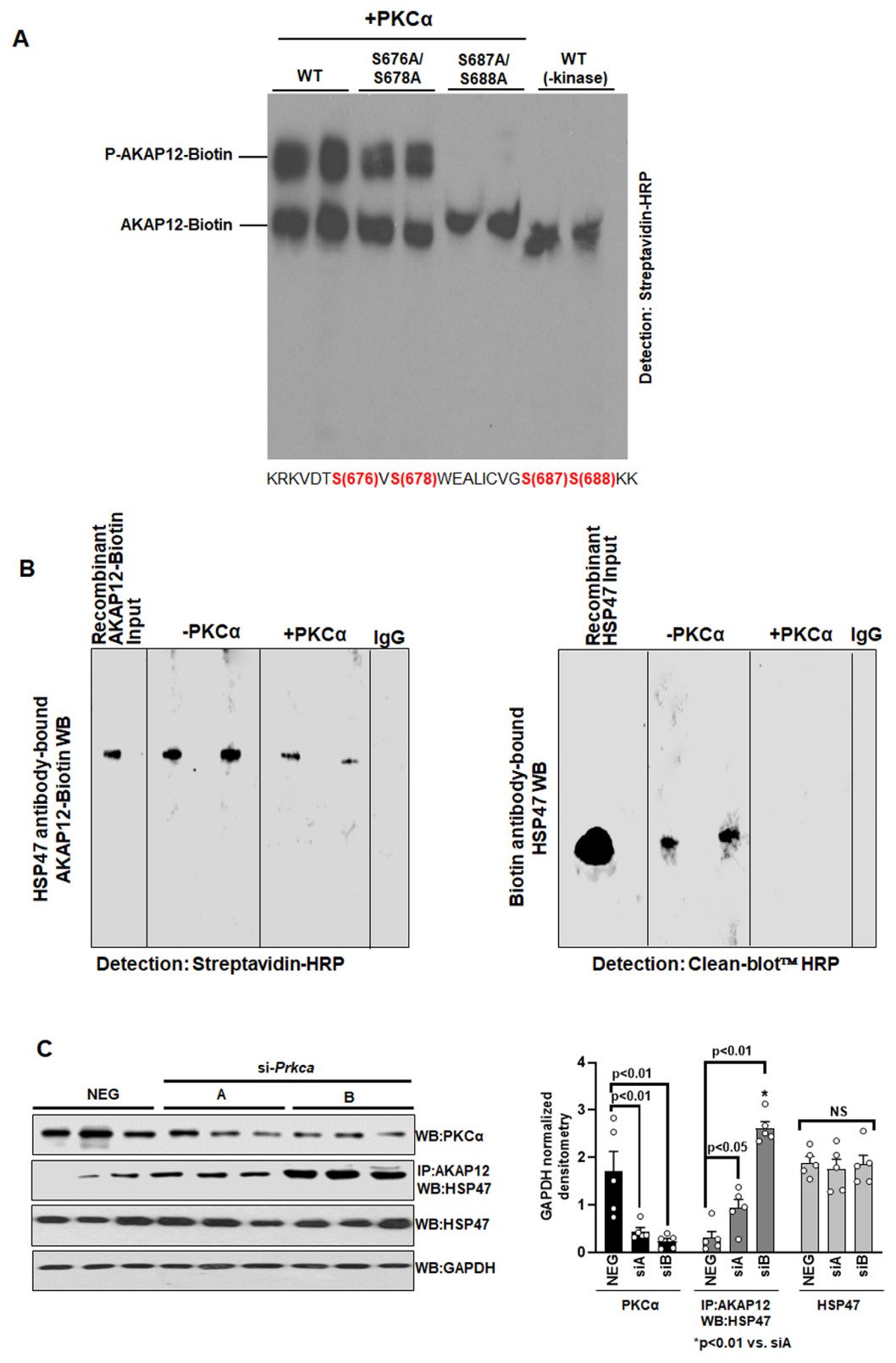

**Figure 3.** PKCα phosphorylates AKAP12 and inhibits its interaction with HSP47. (**A**) AKAP12 is phosphorylated by PKCα at its activation-responsive phospho-sites. Recombinant WT or AKAP12 phospho-mutants were in vitro translated from their vectors and subjected to in vitro kinase assay in the presence of active PKCα enzyme as in Materials and methods. The reactions were run on a phostag gel to detect phosphorylated AKAP12 or its mutants.

*Figure 3 continued on next page*

*Figure 3 continued*

Representative phostag gels from three experiments are shown. Source data are presented in *Figure 3—source data 1*. P values are calculated in *Figure 3—source data 4*. (**B**) Direct Interaction between AKAP12 and HSP47 in recombinant system in the absence or presence of active PKCα enzyme. In vitro translated biotinylated AKAP12 was incubated with recombinant HSP47 antibody column containing bound HSP47 (left) or recombinant HSP47 was incubated with Biotin antibody column containing bound AKAP12-Biotin (right) in the presence or absence of active PKCα as in Materials and methods. Two representative data out of four experiments are shown. Source data are presented in *Figure 3—source data 2*. (**C**) Silencing *Prkca* in activated human HSCs enhances AKAP12-HSP47 interaction. Culture-activated human HSCs were transfected with a universal negative control siRNA (Neg) or two *Prkca* siRNAs (**A** or **B**) as in Materials and methods. Total protein was assessed for AKAP12-HSP47 co-immunoprecipitation or PKCα, HSP47, and GAPDH immunoblotting. Data represented as GAPDH normalized densitometry are mean ± SE from five experiments. Source data are presented in *Figure 3—source data 3*. P values are calculated in *Figure 3—source data 4*. HSC, hepatic stellate cell; WT, wild-type.

The online version of this article includes the following source data for figure 3:

**Source data 1.** Source blots for *Figure 3A*.

**Source data 2.** Source blots for *Figure 3B*.

**Source data 3.** Source blots for *Figure 3C*.

**Source data 4.** Post hoc analysis for *Figure 3C*.

*Figure 4—source data 1*). PCR with deletion-specific primers did not amplify the 256 or 154 bp mutant region in hepatocytes, indicating that CRISPR-editing using an HSC promoter-specific SaCas9 occurred in HSCs but not hepatocytes (*Figure 4E*, *Figure 4—source data 2*). A specific primer to detect PMUT could not be designed, hence PMUT specificity was evaluated by next-generation amplicon sequencing (NGS) as in *Figure 4G*. The efficiency of AAV transduction by PDEL or PMUT donor was evaluated under oil or $CCl_4$ conditions by immunostaining of SaCas9 enzyme with HSC marker, desmin or hepatocyte marker, and albumin. The GFAP-driven SaCas9 enzyme strongly co-localized with desmin-positive HSCs in PDEL or PMUT transduced livers (*Figure 4D*) but not with albumin-positive hepatocytes in the liver (*Figure 4F*). The efficiency of transduction of PDEL or PMUT as calculated by the SaCas9 count per desmin area was not significantly different between PDEL and PMUT under either oil or $CCl_4$ conditions (*Figure 4D*). $CCl_4$ exposure increased the overall numbers of desmin-positive HSCs (*Figure 4D*) due to increased activation and proliferation (*Fujii et al., 2010*). Post hoc analysis of *Figure 4D and F* is provided in *Figure 4—source data 3*. The efficiency of gene editing was tested by using a 298-bp amplicon from HSCs or hepatocytes of GFAP-SaCas9 CRISPR livers (*Figure 4G*). On-target and off-target base changes were analyzed by comparing the target read sequences to the reference sequence of WT *Akap12* amplicon as described under Materials and methods. For the PDEL CRISPR, oil+CR HSCs exhibited 30% mutant reads compared to the total reads whereas $CCl_4$+CR HSCs exhibited 60% mutant reads compared to total (*Figure 4G*, top panel, *Supplementary file 2*). Oil+EV or $CCl_4$+EV HSCs did not contain any mutant reads (*Figure 4G*, top panel). For the PMUT CRISPR, oil+CR HSC exhibited 3% mutant reads and $CCl_4$+CR exhibited 12.5% mutant reads compared to total reads (*Figure 4G*, bottom panel, *Supplementary file 3*). Hepatocytes from the CR groups did not exhibit any PDEL or PMUT sequence reads (*Figure 4G*, *Supplementary files 2 and 3*). The percentage of base changes outside the target region between the two sgRNA sites (*Figure 4A*) was less than 5% in most cases (*Figure 4G*). Like GFAP-SaCas9, PDEL CRISPR was also observed with LRAT-SaCas9 in HSCs (*Figure 4—figure supplement 1*) but not hepatocytes (*Figure 4—figure supplement 1*). The LRAT-driven SaCas9 expression in desmin-positive HSCs was lower in the $CCl_4$ group compared to oil (*Figure 4—figure supplement 1*) and not significant in hepatocytes (*Figure 4—figure supplement 1*). The CRISPR deletion efficiency using LRAT-SaCas9 in HSCs of oil+CR group was 45% whereas that of the $CCL_4$+CR group was 30% of total reads (*Figure 4—figure supplement 1*, *Supplementary file 2*).

## Phospho-editing of AKAP12 regulates liver injury and fibrosis in the $CCl_4$ mouse model

At gross level, $CCl_4$ administration for five weeks reduced the body weight of mice by 20% (*Figure 5A*) and increased the liver to body weight ratio by 1.25-fold compared to oil (*Figure 5B*). AKAP12 phospho-editing by GFAP-SaCas9 in normal mice (oil+CR) did not alter body or liver

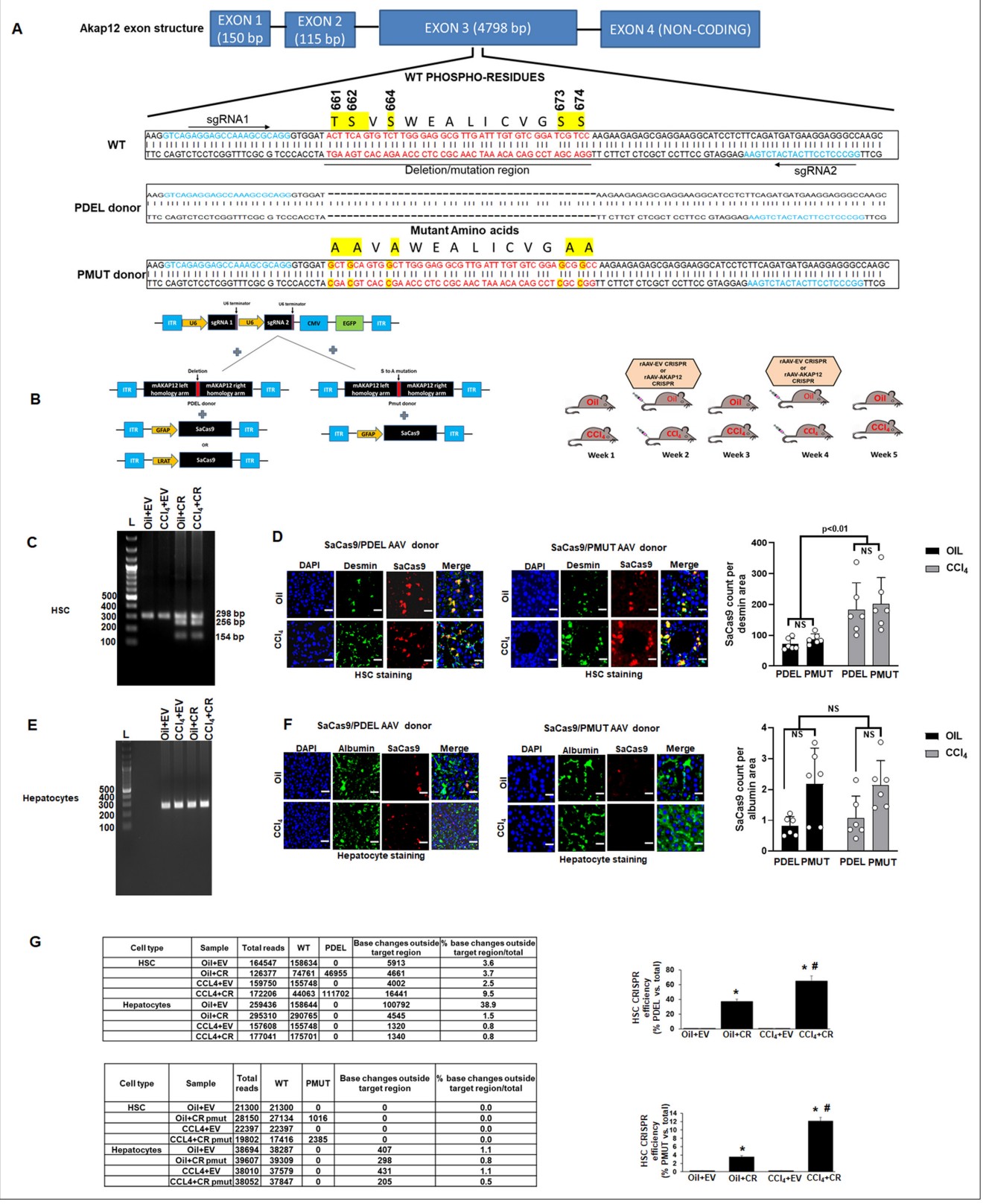

**Figure 4.** In vivo gene editing of the *Akap12* region corresponding to its activation-responsive phospho-sites in the CCL4 mouse model using GFAP-SaCas9. (**A**) Schematic diagram of the mouse *Akap12* locus showing the exon 3 region containing AKAP12's activation-responsive phospho-site regions and two SaCas9 target sgRNAs (1 and 2). The PDEL mutation contains a 42 bp deletion in the donor that deletes the phospho-sites after CRISPR editing. The PMUT donor has a mutation in five codons that changes the S/T (serine/threonine) activation-responsive phospho-sites to A (alanine). (**B**) Left panel:

*Figure 4 continued on next page*

*Figure 4 continued*

AAV-CRISPR cloning scheme. SgRNA1/2, PDEL or PMUT donor and SaCas9 under control of the HSC-specific GFAP promoter were cloned into AAV6 serotype vectors and AAV particles were generated as in methods. Control vector (EV) contained a non-targeting sgRNA as in methods. Combinations of sgRNA1/2-AAV or EV with the pDEL or pMUT donor-AAV and GFAP-SaCas9-AAV were injected into mice as in Materials and methods. Right panel: Scheme of AAV vector injections into the tail vein at second and fourth weeks of oil or CCL4 administration as in methods. (C) Specificity of PDEL CRISPR for HSCs was evaluated by multiplex PCR amplification of genomic DNA from HSCs using a PDEL-specific primer and two primers around the PDEL primer site (see Key resource table). A representative gel image from six experimental groups is shown. Source data are presented in *Figure 4—source data 1*. (D) Efficiency of PDEL (left) or PMUT (right) AAV in HSCs was ascertained by co-localization of SaCas9 with desmin-positive HSCs in oil or CCL4-treated groups. Data represented by SaCas9 count per desmin area are mean ± SE from six experimental groups. 200× magnification; scale bar=50 µm. P values are calculated in *Figure 4—source data 3*. (E) Multiplex PCR of hepatocytes genomic DNA as in (C) above did not show PDEL specific amplicons. A representative gel image from six experiments is shown. Source data are presented in *Figure 4—source data 2*. (F) SaCas9 co-localization with albumin-positive hepatocytes was insignificant in oil or CCL4 livers transduced with PDEL or PMUT AAV compared to HSCs in (D) above. Data represented by fluorescence signal count are mean±SE from six experimental groups. 200× magnification; scale bar=50 µm. P values are calculated in *Figure 4—source data 3*. (G) The efficiency of CRISR was evaluated by NGS using a 298-bp PCR amplicon derived from genomic DNA of HSCs or hepatocytes of PDEL mice group (top panel) or PMUT mice group (bottom panel). Total amplicon reads, WT reads, and PDEL or PMUT reads within the target region or base changes outside the target region from each experimental group are shown. The CRISPR editing efficiency is the represented by the percentage of mutant reads versus total. Oil+CR or CCL4+CR-PDEL/PMUT: *p<0.01 versus oil+EV; #p<0.01 versus CCL4+EV. EV, empty vector; HSC, hepatic stellate cell; WT, wild-type.

The online version of this article includes the following source data and figure supplement(s) for figure 4:

**Source data 1.** Original gel for *Figure 4C*.

**Source data 2.** Original gel for *Figure 4E*.

**Source data 3.** Post hoc analysis for *Figure 4D, F*.

**Figure supplement 1.** In vivo gene editing of the *Akap12* region corresponding to its activation-responsive phospho-sites in the CCl₄ mouse model using LRAT-Cas9.

**Figure supplement 1—source data 1.** Original gel of *Figure 4—figure supplement 1A* .

**Figure supplement 1—source data 2.** Original gel of *Figure 4—figure supplement 1C*.

weight compared to oil+EV (*Figure 5A*). However, AKAP12 phospho-editing in CCL₄ mice (CCl₄+CR) normalized CCL₄+EV-mediated alterations in body weight and liver/body weight to that of oil+EV levels (*Figure 5A*, *Figure 5B*). Histologically, control mice (oil) had a normal hepatic cord pattern around the central vein, whereas fatty vacuolar changes and disorganized hepatic lobular structure with centrilobular fibrosis were observed in CCl₄ livers (*Figure 5C*, PDEL-top panel, PMUT-bottom panel) as referenced previously (*Wang et al., 2012*). AKAP12 phospho-editing by PDEL in CCl₄ mice (CCl4+CR PDEL) dramatically reduced the CCl₄-induced histological distortions compared to CCl₄+EV (*Figure 5C*, top panel). AKAP12 PMUT editing also suppressed the histological changes but less dramatically compared to PDEL (*Figure 5C*, bottom panel), hematoxylin and eosin (H&E) staining for individual PDEL and PMUT experiments is shown in *Figure 5—source data 1*. Picosirus red staining of CCl₄ livers showed increased collagen deposition that was substantially reduced when mice were administered AKAP12 phospho-editing vectors (*Figure 5D*). Sirius red staining for individual PDEL and PMUT experiments is shown in *Figure 5—source data 2*. The hydroxyproline content of collagen was increased 2.4-fold in CCl₄ livers compared to oil+EV and normalized by AKAP12 PDEL (*Figure 5E*, top panel). PMUT phospho-editing inhibited CCl₄-mediated induction but did not completely normalize hydroxyproline content compared to oil+EV or oil+CR (*Figure 5E*, bottom panel). CCl₄ administration caused an 8- to 13-fold induction in liver injury as measured by ALT/AST levels (*Figure 5F*). AKAP12 phospho-editing by PDEL or PMUT in control mice (oil+CR) did not affect the levels of ALT/AST (*Figure 5F*). AKAP12 phospho-editing by PDEL or PMUT in CCl₄ mice (CCl4+CR PDEL or PMUT, left and right panels, respectively) dramatically reduced the ALT/AST level by 75–80% compared to CCl₄+EV (*Figure 5F*). Compared to CCl₄+CR PDEL, CCl₄+CR PMUT ALT levels were not normalized to oil+CR levels but were statistically higher than that of oil+CR group (*Figure 5F*, right panel). Post hoc analysis of *Figure 5A, B, E and F* is presented as *Figure 5—source data 3*. LRAT-SaCas9 directed PDEL-CRISPR also resulted in higher body weight, lower liver/body weight ratio, and suppression of CCl₄-induced histological changes like that of GFAP-SaCas9.

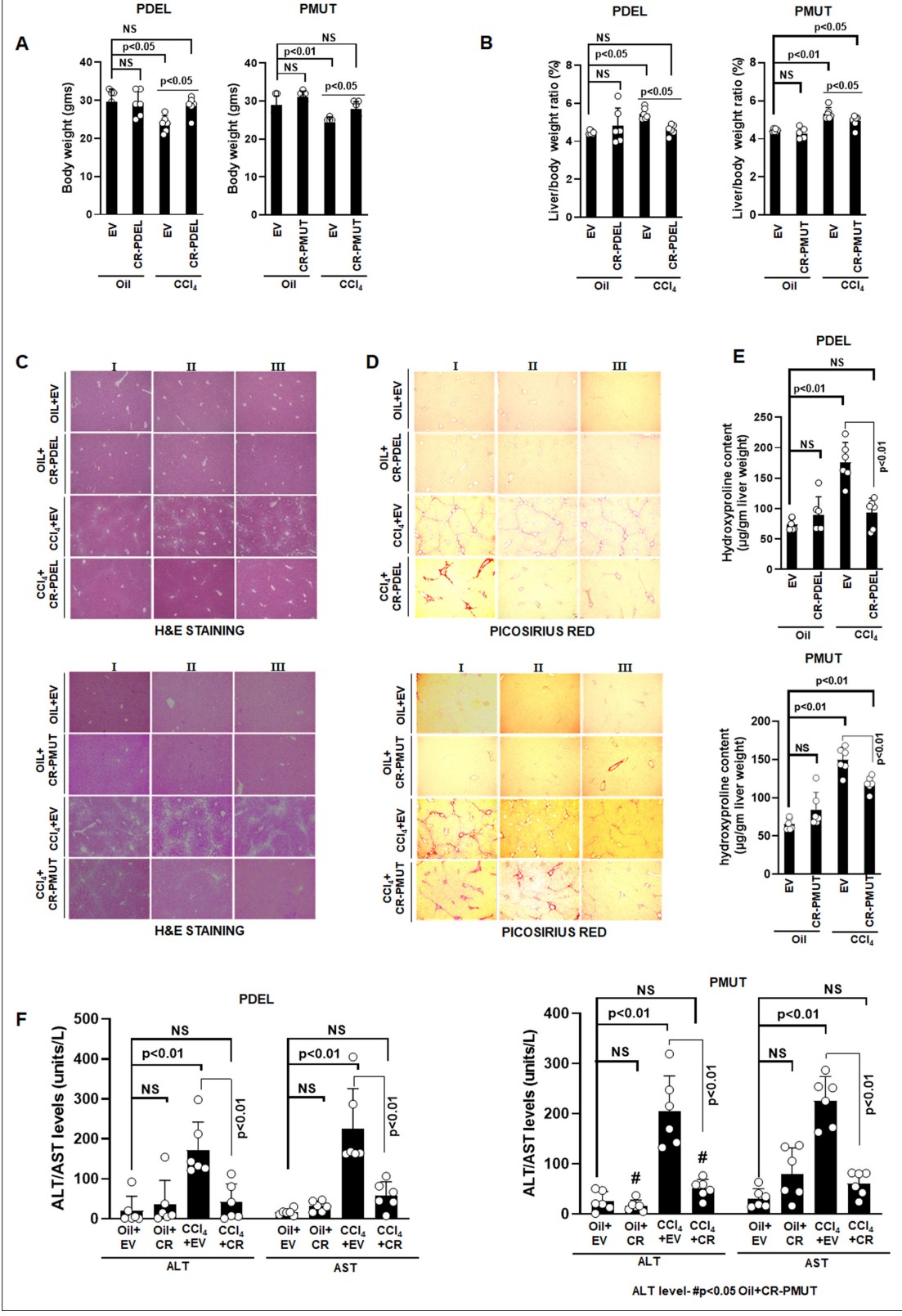

**Figure 5.** Phospho-editing of AKAP12 regulates liver injury and fibrosis in the CCL₄ mouse model. Gross changes in mouse body weight (**A**) and liver/body weight ratio (**B**) after PDEL or PMUT GFAP-SaCas9-mediated CRISPR editing of AKAP12's phospho-sites under oil or CCL₄ treatment conditions. Mean±SE from six PDEL or PMUT experiments. P values are calculated in *Figure 5—source data 3*. (**C**) Histological evaluation of CRISPR-edited livers by H&E staining as in methods for six PDEL (top) or PMUT (bottom) experiments. Source data are presented in *Figure 5—source data 1*. (**D**) Picosirius

*Figure 5 continued on next page*

*Figure 5 continued*

red staining of CRISPR-edited livers for collagen from six PDEL (top) or PMUT (bottom) experiments. Source data are presented in *Figure 5—source data 2*. (**E**) Hydroxyproline quantification of collagen (mean±SE) from six PDEL (top) or PMUT experiments. P values are calculated in *Figure 5—source data 3*. (**F**) Measurement of ALT and AST levels in plasma after CRISPR-editing as in Materials and methods. Mean±SE from six PDEL (left panel) or PMUT (right panel) experiments. P values are calculated in *Figure 5—source data 3*. H&E, hematoxylin and eosin.

The online version of this article includes the following source data and figure supplement(s) for figure 5:

**Source data 1.** Individual images for *Figure 5C*.

**Source data 2.** Individual images for *Figure 5D*.

**Source data 3.** Post hoc analysis for *Figure 5*.

**Figure supplement 1.** Phospho-editing of AKAP12 using LRAT-Cas9 regulates liver fibrosis in the CCl4 mouse model.

## Phospho-editing of AKAP12 regulates AKAP12's HSP47-scaffolding activity, HSC activation, and HSP47's collagen-chaperoning activity in the CCl$_4$ mouse model

The AKAP12-HSP47 scaffold was reduced in livers of CCl$_4$+EV mice compared to oil controls (*Figure 6A and B*). PDEL-CRISPR or PMUT-CRISPR editing in CCl$_4$ mice restored the drop in the AKAP12-HSP47 interaction caused by CCl$_4$ (*Figure 6A*, PDEL; *Figure 6B*, PMUT, *Figure 6—source data 1*, PDEL; *Figure 6—source data 2*, PMUT). AKAP12 phospho-editing dramatically inhibited CCl$_4$-mediated HSC activation as evidenced by a drop in α-SMA levels (*Figure 6A and B*). In conjunction with restoration of AKAP12-HSP47 scaffold, the increased interaction between collagen and HSP47 upon CCl$_4$ exposure was inhibited by AKAP12 phospho-editing (*Figure 6C*). Co-immunoprecipitation of collagen with HSP47 antibody yielded non-specific bands at positions above the collagen position in all samples including IgG control. The original uncropped blot is shown in *Figure 6—source data 3*. AKAP12 PDEL or PMUT phospho-editing also inhibited the increase in *Col1a1* and *Acta2* mRNA levels caused by CCl$_4$ exposure (*Figure 6D*). *Col1a1* levels were normalized to oil+EV levels by both PDEL and PMUT CRISPRs (*Figure 6D*, left and right panels). However, compared to PDEL, PMUT CRISPR reduced but did not completely normalize *Acta2* levels to control (oil+EV) state (*Figure 5D*, left and right panel). PLA staining showed that the AKAP12-HSP47 scaffold was localized with desmin-positive HSCs under normal (oil) conditions (*Figure 6D*). A drop in AKAP12-HSP47-desmin co-localization was observed upon CCl$_4$ exposure that was restored by AKAP12 PDEL or PMUT phospho-editing (*Figure 6D*). Post hoc analysis for *Figure 6A–E* is presented as *Figure 6—source data 4*.

## HSC-specific phospho-editing of AKAP12 regulates the ER stress response

To determine how HSC-specific AKAP12 phospho-editing reduced overall liver injury and modulated collagen mRNA levels upon CCl$_4$ exposure, we performed proteomics analysis of HSCs and livers isolated from oil+EV, oil+CR, CCl$_4$+EV, or CCl$_4$+CR groups to compare the molecular changes under these conditions. Proteomics analysis revealed alterations in several proteins in CCl$_4$ HSCs as well as total liver that were regulated by AKAP12 HSC-specific phospho-editing (*Supplementary file 4*). Ingenuity pathway analysis (IPA) of these proteins identified two top scoring pathways, the ER stress response and UPR, that were significantly dysregulated by CCl$_4$ and were normalized upon AKAP12 phospho-editing (*Supplementary file 4*). The proteomics analysis of HSCs showed an induction in BIP/GRP78, an ER stress sensor (*Sepulveda et al., 2018*), in CCL$_4$-treated group (*Figure 7A*, *Supplementary file 4*). We confirmed the proteomics by western blotting. GAPDH-normalized densitometries from individual experiments are presented in *Figure 7—figure supplement 1*. HSCs isolated from CCl$_4$ livers showed increased BIP expression (*Figure 7B*). However, even though the proteomics analysis showed inhibition of CCl$_4$-induced BIP levels by AKAP12 phospho-editing (*Figure 7A*), we could not confirm this effect by western blotting (*Figure 7B*). Since BIP is a known collagen chaperone (*Sepulveda et al., 2018*), we examined its interaction with collagen in our CRISPR model. BIP exhibited increased interaction with collagen in the CCl$_4$+EV HSCs compared to oil+EV HSCs and AKAP12 phospho-editing strongly inhibited the BIP-collagen interaction in HSCs (*Figure 7B*). IRE1α, a UPR component that binds to HSP47 and becomes phosphorylated during ER stress (*Sepulveda et al., 2018*), exhibited increased interaction with HSP47 in CCL$_4$ HSCs that was inhibited by

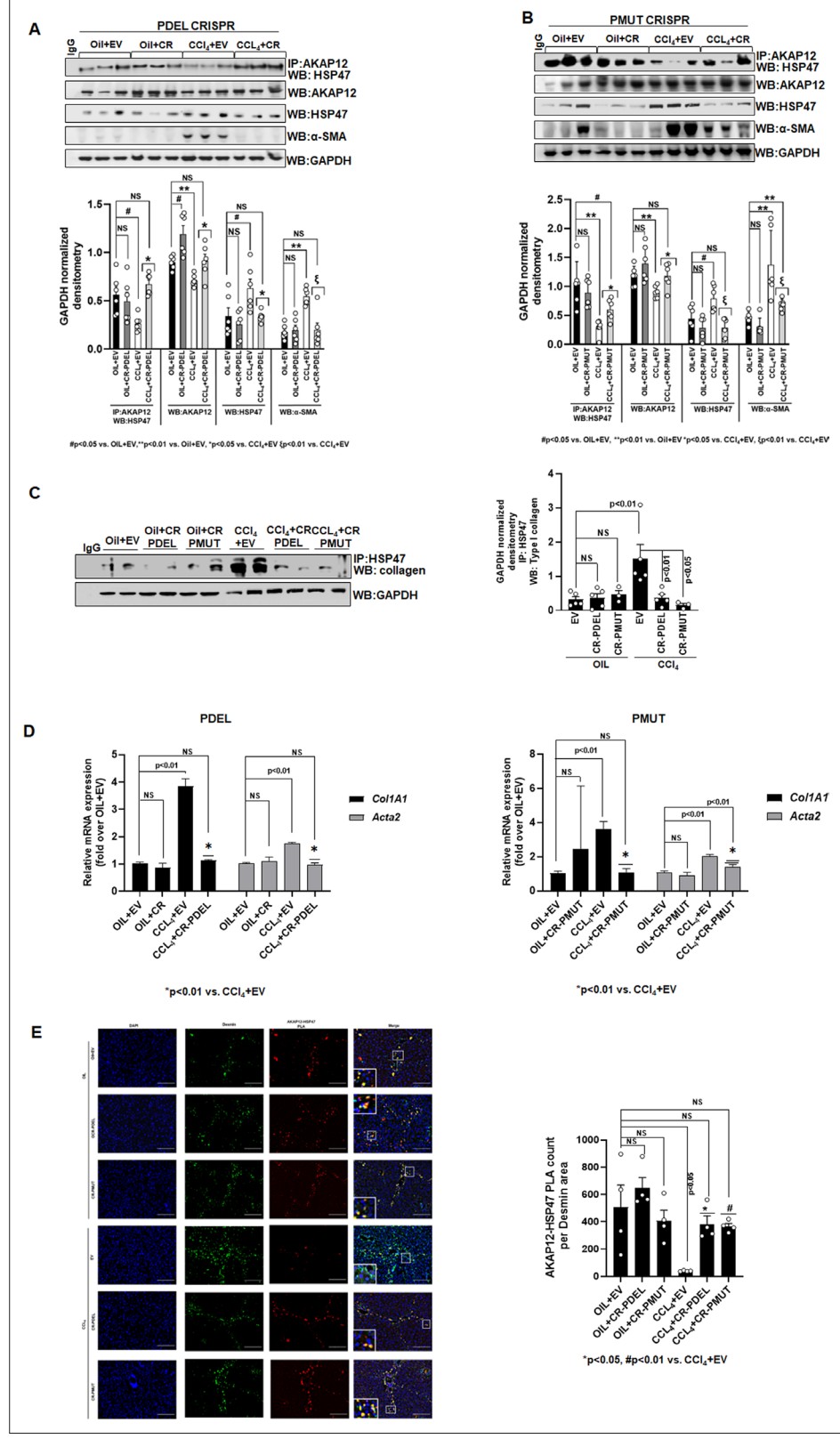

**Figure 6.** Phospho-editing of AKAP12 regulates AKAP12's HSP47-scaffolding activity, HSC activation, and HSP47's collagen-chaperoning activity in the $CCl_4$ mouse model. (**A**) AKAP12-HSP47 co-immunoprecipitation, AKAP12, HSP47, and α-SMA western blotting from liver protein of CR-PDEL experiment. Data represented as GAPDH normalized densitometry are mean±SE from six experiments. Three representatives are shown. Source data are

*Figure 6 continued on next page*

*Figure 6 continued*

presented in *Figure 6—source data 1*. P values are calculated in *Figure 6—source data 4*. (**B**) AKAP12-HSP47 co-immunoprecipitation, AKAP12, HSP47, and α-SMA western blotting from liver protein of CR-PMUT experiment. Data represented as GAPDH normalized densitometry are mean±SE from six experiments. Three representatives are shown. Source data are presented in *Figure 6—source data 1*. P values are calculated in *Figure 6—source data 4*. (**C**) Co-immunoprecipitation of collagen with HSP47 in CRISPR-edited livers are mean±SE from five PDEL and three PMUT experiments. Source data are presented in *Figure 6—source data 3*. P values are calculated in *Figure 6—source data 4*. (**D**) *Col1A1* or *Acta2* mRNA levels by real-time PCR from PDEL (left) or PMUT (right) mouse livers. Data are mean±SE from six experimental groups. P values are calculated in *Figure 6—source data 4*. (**E**) Interaction between AKAP12 and HSP47 in desmin-positive HSCs of CRISPR model by PLA staining. Data representative of the AKAP12-HSP47 PLA count per desmin area are mean±SE from four PDEL or PMUT experiments. 200× magnification, scale bar=100 μm. P values are calculated in *Figure 6—source data 4*. HSC, hepatic stellate cell; PLA, proximity ligation assay.

The online version of this article includes the following source data for figure 6:

**Source data 1.** Source blots for *Figure 6A*.

**Source data 2.** Source blots for *Figure 6B*.

**Source data 3.** Source blots for *Figure 6C*.

**Source data 4.** Post hoc analysis for *Figure 6*.

---

AKAP12 phospho-editing (*Figure 7B*). The IRE1α-HSP47 interaction was further confirmed in desmin-positive HSCs of the CRISPR model by PLA staining (*Figure 7—figure supplement 2*). CCl₄-mediated IRE1α phospho-activation (S724 phosphorylation) was strongly inhibited by AKAP12 phospho-editing without a change in total IRE1α levels (*Figure 7B*). Furthermore, two pathways, P38MAPK and SMAD2/3 that are known to be induced in HSCs through IRE1α activation (*de Galarreta et al., 2016*) were also suppressed by AKAP12 phospho-editing (*Figure 7B*). The proteome of CCl₄-exposed livers exhibited increased ER stress and UPR signaling components that were modulated by AKAP12 HSC-specific phospho-editing (*Figure 7A*, *Supplementary file 4*). BIP levels by western blotting were induced in CCl₄ livers and inhibited by AKAP12 phospho-editing, confirming the proteomics result (*Figure 7C*). Like the proteomics data, we did not find any change in total IRE1α expression. However, phospho-activated IRE1α was suppressed by AKAP12 phospho-editing in total liver (*Figure 7C*). Since ER stress induces inflammatory signals in different systems (*Maiers and Malhi, 2019*), we examined whether the HSCs from our CRISPR mouse model exhibited altered inflammatory signaling upon AKAP12 phospho-modulation. Out of the known HSC cytokines, we found the pro-inflammatory cytokine, IL-17, IL-6, and IL-β to be strongly induced in CCl₄-HSCs whereas AKAP12-phospho-edited HSCs suppressed their expression (*Figure 7D*). On the other hand, an anti-inflammatory cytokine, IL-10 was suppressed in HSCs by CCl₄ administration, and its expression was restored by AKAP12 phospho-editing (*Figure 7D*). To examine whether ER stress modulation within activated HSCs was transmitted to other liver cell types, we evaluated crosstalk between HSCs and hepatocytes in a co-culture system where AKAP12 was CRISPR-edited. Co-culture with activated HSCs induced the ER stress response markers BIP and induced IRE1α phosphorylation in hepatocytes compared to co-culture with quiescent HSCs (*Figure 7E*). Co-culture with activated HSCs in which AKAP12 was phospho-edited (CR) reduced the ER stress signal in hepatocytes compared to activated HSCs alone whereas hepatocytes co-cultured with quiescent HSCs with CR did not exhibit any alteration in ER stress markers compared to WT (*Figure 7E*). Original blots for *Figure 7B–E* are shown in *Figure 7—source data 1* and *Figure 7—source data 4*. Post hoc analysis for *Figure 7B–E* is presented in *Figure 7—source data 5*.

## Discussion

In fibrotic mouse and human livers, HSCs exhibit increased AKAP12 phosphorylation and decreased AKAP12 scaffolding activity toward the collagen chaperone, HSP47. By mapping the phosphorylation events that are altered upon activation of human or mouse HSCs, we have demonstrated that phosphorylation of specific S or T residues of AKAP12 is triggered during HSC activation. Hence, we named these sites as activation-responsive phospho-sites. Out of the five activation-responsive phospho-sites, four serine residues were confirmed as PKCα substrates. Mutagenesis analysis on recombinant AKAP12 showed that the S687 and S688 were stronger PKCα substrates compared to S676 and S678

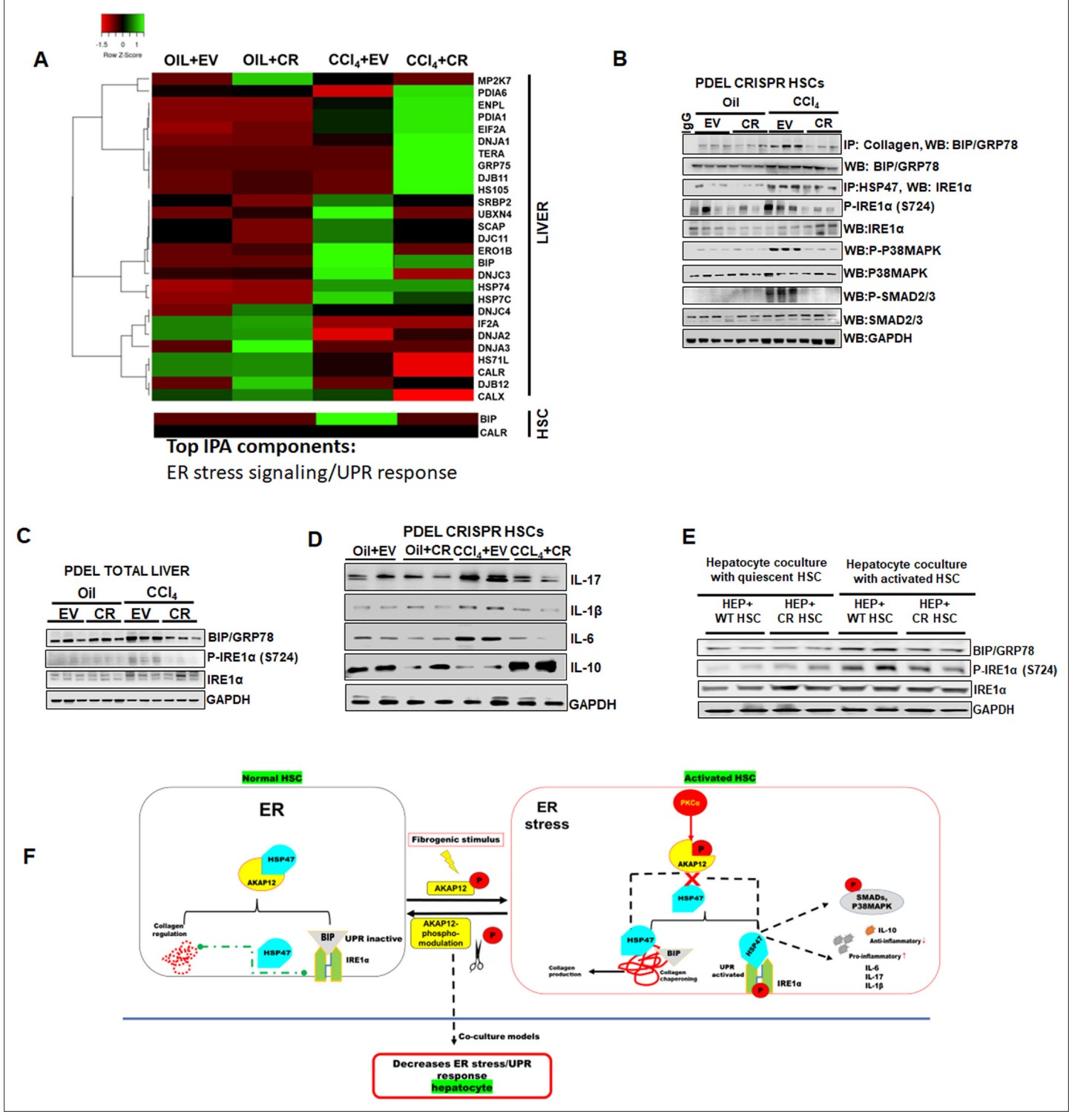

**Figure 7.** HSC-specific phospho-editing of AKAP12 regulates the ER stress response. (**A**) Heat map of total liver and HSCs ER stress/UPR signaling components in four groups, oil+EV, oil+CR, CCl$_4$+EV and CCL$_4$+CR. Proteomics data utilized to prepare the heatmap are presented in **Supplementary file 4**. Alterations in ER stress responsive elements in HSCs from AKAP12 PDEL CRISPR model (**B**) and total liver (**C**). GAPDH normalized densitometry is mean±SE from three experiments. Raw densitometry of each experiment is presented in **Figure 7—figure supplement 1**. Source data are presented in **Figure 7—source data 1** and **Figure 7—source data 2**. P values are calculated in **Figure 7—source data 5**. (**D**) Inflammatory cytokine expression in HSCs from oil+EV, oil+CR, CCl$_4$+EV and CCL$_4$+CR groups. GAPDH normalized densitometry is mean±SE from four experiments. Raw densitometry of each experiment is presented in **Figure 7—figure supplement 1**. Source data are presented in **Figure 7—source data 3**. P values are calculated in

*Figure 7 continued on next page*

*Figure 7 continued*

*Figure 7—source data 5*. (**E**) ER stress response in hepatocytes co-cultured with quiescent or activated WT HSCs with or without AKAP12 PDEL CRISPR editing (CR). GAPDH normalized densitometry represented as fold over hepatocytes +WT quiescent HSCs is mean±SE from three experiments. Raw densitometry of each experiment is presented in *Figure 7—figure supplement 1*. Source data are presented in *Figure 7—source data 4*. P values are calculated in *Figure 7—source data 5*. (**F**) Summary of findings. AKAP12 interacts with HSP47 in the ER of normal HSCs and negatively regulates HSP47's collagen-chaperoning activity and its ability to promote ER stress-directed IRE1α branch of UPR signaling. AKAP12 negatively regulates HSC activation. Pro-fibrogenic stimuli that cause HSC activation allow AKAP12's PKCα-dependent site-specific phosphorylation. By AKAP12 CRISPR phospho-editing, we show that AKAP12 phosphorylation inhibits AKAP12's HSP47 scaffolding activity, increases HSP47-collagen chaperoning activity and induces HSP47's interaction with UPR signals (IRE1α and P-IRE1α). AKAP12 phosphorylation may lead to increased downstream events associated with the UPR signaling such as BIP-collagen chaperoning, phosphorylation of SMADs/P38MAPK and UPR-regulated inflammatory signaling. Blocking AKAP12 phosphorylation in activated HSCs prevents the ER stress response in hepatocytes co-cultured with activated HSCs. ER, endoplasmic reticulum; HSC, hepatic stellate cell; UPR, unfolded protein response; WT, wild-type.

The online version of this article includes the following source data and figure supplement(s) for figure 7:

**Source data 1.** Source blots for *Figure 7B*.

**Source data 2.** Source blots for *Figure 7C*.

**Source data 3.** Source blots for *Figure 7D*.

**Source data 4.** Source blots for *Figure 7E*.

**Source data 5.** Post hoc analysis for *Figure 7*.

**Figure supplement 1.** Densitometric quantification of blots from *Figure 7B–E*.

**Figure supplement 2.** Interaction between IRE1α and HSP47 in desmin-positive HSCs of CRISPR-PDEL model by PLA staining.

because their mutations drastically suppressed phosphorylation. We further observed that phosphorylation of AKAP12 by PKCα suppressed direct binding between AKAP12 and HSP47. Confirming this recombinant data, we observed that silencing *PKCα* in HSCs induced the binding between AKAP12 and HSP47. The data suggest that PKCα is involved in promoting AKAP12's scaffolding activity toward HSP47. AKAP12's known scaffolding activities toward CCND1, PLK1, and PKCα that were identified previously are regulated by its phosphorylation (*Canton et al., 2012*; *Akakura and Gelman, 2012*; *Burnworth et al., 2012*). We therefore evaluated the role of site-specific phosphorylation in modulating AKAP12's scaffolding functions in HSCs.

Using a CRISPR-based gene editing approach, we deleted AKAP12's phosphorylation sites in culture-activated human or mouse HSCs and observed an enhancement in AKAP12's interaction with HSP47, a strong inhibition of HSC activation (judged by α-SMA levels) and restoration of the quiescent marker, vitamin A that is suppressed in activated HSCs (*Senoo et al., 2010*). HSP47 resides in the ER (*Kawasaki et al., 2015*) and since AKAP12 interacted with HSP47, we evaluated whether it co-localized with HSP47 in the ER and whether the AKAP12-HSP47 scaffold in the ER was affected by CRISPR-editing its phosphorylation sites. The AKAP12-HSP47 scaffold was induced in the ER upon AKAP12 phospho-editing. HSP47's chaperoning activity toward collagen is highly induced during HSC activation and this allows increased maturation and secretion of collagen (*Kawasaki et al., 2015*). Since AKAP12 binds to HSP47, we examined whether this interaction regulated HSP47's collagen chaperoning function. The ER of activated HSCs stained strongly for the collagen-HSP47 scaffold but AKAP12 phospho-site editing diminished the collagen scaffolding activity of HSP47. Our findings suggest that lack of AKAP12 activation-responsive phosphorylation quenches HSP47's collagen chaperoning activity and prevents HSC activation.

HSC activation is a hallmark of liver fibrosis. The fact that enhanced phosphorylation of AKAP12 at its activation-responsive phospho-sites promotes HSC activation fueled our hypothesis that site-specific AKAP12 phosphorylation may be involved in promoting liver fibrosis in animal models. To address this hypothesis, we designed CRISPR-AAV vectors to perform gene editing of AKAP12's activation-responsive phospho-sites specifically in HSCs of mouse liver. This was achieved by expressing the CRISPR-causing enzyme, SaCas9 under control of the HSC-specific promoters, GFAP or LRAT (*Puche et al., 2013*; *Lee et al., 2020*). Both GFAP and LRAT specifically expressed SaCas9 in HSCs, but GFAP-driven SaCas9 was increased in activated HSCs compared to normal HSCs whereas the reverse was observed with LRAT-Cas9. GFAP promoter activity is induced during HSC activation (*Maubach et al., 2006*) whereas LRAT expression is known to be suppressed (*Khomich et al., 2019*). This might have been responsible for the different effects of these two promoters.

The AAV particles of serotype 6 were used because AAV6 efficiently transduces activated HSCs in the CCl$_4$ mouse model (**Nakano et al., 2020**). HSC-specific gene editing of AKAP12 was performed by deleting the DNA region corresponding to the five phospho-sites (PDEL). AKAP12 phospho-site editing by this PDEL mechanism strongly inhibited HSC activation, enhanced the AKAP12-HSP47 scaffold, and suppressed the collagen-chaperoning activity of HSP47 leading to decreased collagen production in the liver. To confirm the involvement of AKAP12 phosphorylation at these residues in promoting pro-fibrogenic phenotype, we inhibited phosphorylation at these sites by CRISPR-mediated editing of the S/T residues to A (PMUT). The overall editing efficiency caused by PMUT was lower than that of PDEL in activated HSCs from CCl$_4$-exposed livers. PMUT suppressed but did not completely normalize ALT levels and hydroxyproline content compared to PDEL. Despite these differences, PMUT was effective in suppressing the fibrogenic response in the liver, supporting an important role of AKAP12 phosphorylation in regulating the outcome of liver fibrosis. As opposed to fibrotic livers, CRISPR editing in HSCs of normal liver did not alter the molecular identity of the liver. Since normal HSCs do not exhibit phosphorylation of AKAP12 at the activation-responsive phospho-sites, they appear to be unaffected by modulating these sites. This control data reiterates the fact that increased AKAP12 phosphorylation caused by HSC activation has pro-fibrogenic effects.

Apart from suppression of fibrotic parameters, we observed that AKAP12 phospho-modulation in HSCs inhibited collagen mRNA levels and globally suppressed liver injury. Since inhibition of collagen transcription and overall liver injury may not be due to AKAP12's ability to regulate HSP47's collagen-chaperoning activity, we searched for additional mechanisms of action of phospho-AKAP12. We performed proteomics analysis in HSCs from our CRISPR model and total liver from the same to identify molecular signals altered by AKAP12 phospho-editing. In HSCs, we identified BIP/GRP78, a regulator of the IRE1α branch of UPR signaling and a known collagen chaperone in the ER (**Sepulveda et al., 2018**; **Ferreira et al., 1994**). Interestingly, a recent interactome study identified HSP47 as a binding partner for IRE1α (**Sepulveda et al., 2018**). IRE1α is an ER transmembrane kinase that is kept in inactive state by its binding to BIP. HSP47 activates IRE1α oligomerization and phosphorylation by displacing BIP and triggering the UPR response during ER stress (**Sepulveda et al., 2018**). The functional effect of the HSP47-IRE1α interaction on UPR signaling and collagen folding during fibrogenic stimulation in HSCs is undescribed so far. But IRE1α activation caused by ER stress inducers in HSCs is known to enhance collagen transcription as well as collagen protein expression through activation of p38MAPK and SMAD pathways (**de Galarreta et al., 2016**). HSCs exhibit ER stress and UPR signaling in response to liver injury stimuli (**Maiers and Malhi, 2019**; **Mannaerts et al., 2019**). In fact, ER stress appears to be both a cause and effect of HSC activation (**Maiers and Malhi, 2019**; **Koo et al., 2016**). Since phospho-edited AKAP12 interacted with HSP47 in the ER of HSCs, we wondered whether HSP47-mediated UPR signaling might be regulated by AKAP12. We found that IRE1α-HSP47 interaction (**Sepulveda et al., 2018**) was enhanced in CCl$_4$ HSCs and so were downstream pathways known to be enhanced by IRE1α activation in HSCs (phospho-P38MAPK and SMAD2/3) (**de Galarreta et al., 2016**). Interestingly AKAP12 phospho-editing suppressed HSP47's UPR-activating function by quenching the CCl$_4$-mediated IRE1α-HSP47 interaction in HSCs that further inhibited IRE1α phospho-activation preventing downstream P38MAPK and SMAD signaling in these cells. Another component of the UPR signaling we found from proteomics was BIP. BIP is a collagen chaperone that also inactivates IRE1α under basal conditions (**Sepulveda et al., 2018**). During ER stress, HSP47 displaces BIP from IRE1α, activating IRE1α-mediated UPR signaling (**Sepulveda et al., 2018**). Whether this HSP47-mediated BIP displacement promotes BIP's activity as a collagen chaperone during HSC activation or liver fibrosis is unclear so far. We hypothesized that AKAP12 by virtue of its increased phosphorylation and lack of scaffolding toward HSP47 may regulate the BIP-IRE1α-HSP47 axis and promote BIP's collagen chaperoning function. Indeed, we observed increased interaction of BIP with collagen in HSCs of CCL$_4$ livers that was suppressed by AKAP12 phospho-editing. We could not find any interaction between AKAP12 and BIP in HSCs but speculate that loss of AKAP12-HSP47 scaffolding leading to increased HSP47-IRE1α interaction might have released BIP from the IRE1α sites and favored BIP-collagen scaffolding.

Enhanced protein secretion is associated with ER stress and UPR signaling in activated HSCs and is crucial for processing of inflammatory proteins and ECM components upon pro-fibrogenic

stimulation (*Maiers and Malhi, 2019*). Studies in liver and other systems support the role of ER stress in promoting inflammatory signaling (*Garg et al., 2012*; *Hasnain et al., 2012*). Also, inflammatory proteins have a less well-described role in promoting ER stress and UPR signaling (*Hasnain et al., 2012*). ER stress is therefore both a cause and consequence of inflammatory signaling (*Hasnain et al., 2012*). Cytokines such as IL-1β are known to be induced in activated HSCs through ER stress (*He et al., 2018*). Other cytokines known to be expressed by HSCs, IL-17 and IL-6 (*Meng et al., 2012*; *Salguero Palacios et al., 2008*), are prone to modulation by ER stress (*Yang et al., 2018*; *Sanchez et al., 2019*). The anti-inflammatory and antifibrotic cytokine, IL-10 expressed by HSCs (*Hung et al., 2005*) was recently shown as a target of ER stress in macrophages (*Hansen et al., 2019*). IL-6 and IL-1β are mediators of ER stress in the liver (*Duvigneau et al., 2019*). In pancreatic beta cells, IL-1β is known to induce ER stress in a nitric oxide-dependent manner (*Kacheva et al., 2011*). The anti-inflammatory effect of IL-10 has been shown to block ER stress in intestinal epithelial cells (*Shkoda et al., 2007*). Since our data on AKAP12 suggests that it regulates ER stress pathways in HSCs, we tested whether known inflammatory signals linked to ER stress were also regulated by AKAP12. We found the pro-inflammatory cytokines, IL-17, Il-1β, and Il-6 to be induced in CCl$_4$-HSCs whereas AKAP12-phospho-edited HSCs exhibited a strong suppression of these cytokines. On the other hand, the anti-inflammatory cytokine, IL-10 was suppressed in HSCs by CCl$_4$ administration, and its expression was restored by AKAP12 phospho-editing. Literature suggests that inflammatory molecules and UPR signaling may contribute to increased collagen transcription during liver fibrosis. Pro-inflammatory IL-6 signaling induces collagen transcription (*Kagan et al., 2017*) and loss of anti-inflammatory signals such as IL-10 inhibit it (*Wang et al., 1998*). The IRE1α-directed UPR also induces collagen transcription through increased p38MAPK and SMAD2/3 signaling (*de Galarreta et al., 2016*). Since AKAP12 phospho-editing suppressed IRE1α-directed UPR signaling through its association with HSP47 and regulated ER stress-linked cytokines expressed in HSCs, these factors may have contributed to the overall drop in collagen mRNA levels.

Since ER stress/UPR signaling plays a role in enhancing liver injury and the ER stress inducer, tunicamycin is known to induce ALT/AST levels (*Han et al., 2016*), we examined whether AKAP12 HSC-specific editing regulated the liver ER stress response. We found dysregulation of ER stress and UPR-associated components in total liver of CCl$_4$ mice (BIP and other ER foldases such as protein disulfide isomerases, PDIA1, and PDIA6) that were regulated by HSC-specific AKAP12 phospho-editing. Induction of BIP expression in the liver was normalized by AKAP12 phospho-editing. Although the total IRE1α levels were unchanged by CCl$_4$, IRE1α phospho-activation was inhibited by AKAP12 HSC-specific phospho-editing. These results suggest that controlling the ER stress response/UPR signaling within HSCs during pro-fibrogenic stimulation also modulates the same in the whole liver. The phenomenon of ER stress being communicated from stressed cells to other cells within a tissue has been reviewed in the context of cells that produce large amounts of proteins such as immune cells (*Garg et al., 2012*). It has also been published that ER stress invokes liver fibrosis primarily due to ER stress within HSCs due to their activation (*Koo et al., 2016*). Since hepatocytes are known to be sensitive to CCl$_4$-mediated ER stress (*Üstüner et al., 2021*), we examined whether crosstalk between activated HSCs and hepatocytes in a co-culture system promoted the ER stress response in hepatocytes and whether AKAP12 regulated this crosstalk. Modulating HSC activation through AKAP12 regulated the ER stress response in hepatocytes in culture. Since we observed regulation of ER stress-linked inflammatory cytokine production from HSCs of AKAP12 CRISPR edited livers, we speculate that inflammatory cytokines from HSCs might transmit ER stress to the whole liver and AKAP12 provides a mode to control these effects during fibrogenesis. The overall findings are summarized in *Figure 7F*.

In summary, we have identified AKAP12 as a scaffolding partner of HSP47 in normal HSCs that controls HSP47's collagen chaperoning activity and its interaction with UPR signals in HSCs. Site-specific phosphorylation of AKAP12 occurs during HSC activation and this modification inhibits its interaction with HSP47. This induces HSP47's collagen chaperoning activity, collagen production, and HSP47's interaction with UPR signaling proteins upon pro-fibrogenic stimulation. Blocking AKAP12 phospho-modification inhibits HSC activation, collagen production, fibrosis as well as overall liver injury possibly via modulation of the ER stress response and inhibition of ER stress-linked inflammatory signals. The next step in this analysis would be to perform structural studies to identify how AKAP12's activation-responsive phospho-sites interact with HSP47 and the kinase, PKCα. This will

further facilitate the design of small molecules to block AKAP12-PKCα interaction at these sites, thereby preventing phosphorylation and promoting AKAP12-HSP47 scaffolding. Since AKAP12 phospho-modification is not evident in normal HSCs but is induced upon pro-fibrogenic stimulation, AKAP12 phosphorylation may be utilized as a druggable target in liver fibrosis.

# Materials and methods

## Key resources table

| Reagent type (species) or resource | Designation | Source or reference | Identifiers | Additional information |
|---|---|---|---|---|
| Gene (Human) | *AKAP12* | GenBank | Accession ID: NM_005100.4 | |
| Gene (*Mus musculus*) | *Akap12* | GenBank | Accession ID: NM_031185.3 | |
| Transfected construct (Human) | *Negative control siRNA* | Thermo Fisher Scientific | *Cat# 4404021* | *silencerselect siRNA* |
| Transfected construct (Human) | *Prkca siRNA-A* | Thermo Fisher Scientific | *Cat# s11092* | *silencerselect siRNA* |
| Transfected construct (Human) | *Prkca siRNA-B* | Thermo Fisher Scientific | *Cat# s11094* | *silencerselect siRNA* |
| Sequence-based reagent | Human PDEL region forward primer-653 bp amplicon | This paper | PCR primer | AGCTACTTCCGATGGAGAGA |
| Sequence-based reagent | Human PDEL region reverse primer-653 bp amplicon | This paper | PCR primer | CAGGAATAAACTTCTTGATTGAGACC |
| Sequence-based reagent | Human PDEL-specific primer | This paper | PCR primer | GACCCTCTCCTTGCTCTTTTCTTATC |
| Sequence-based reagent | Mouse PDEL region forward primer-780 bp amplicon | This paper | PCR primer | GATGAAGAGCCAGGAGAATACC |
| Sequence-based reagent | Mouse PDEL region reverse primer-780 bp amplicon | This paper | PCR primer | GGAAACCCAAGATTCCTCTCTAC |
| Sequence-based reagent | Mouse PDEL region amplicon sequencing forward primer-298 bp amplicon | This paper | PCR primer | ACAAGGAAGAAGAGCTGGATAAG |
| Sequence-based reagent | Mouse PDEL region amplicon sequencing reverse primer-298 bp amplicon | This paper | PCR primer | CTGGCAGGAAGAGCATCTG |
| Sequence-based reagent | Mouse PDEL -specific primer | This paper | PCR primer | GCCTTCCTCGCTCTCTTCTTATC |
| Sequence-based reagent | Human guide sequence | This paper | CRISPR guide RNA sequence | GGAAGAACCAAAGCGCAAGGTG |
| Sequence-based reagent | Mouse guide sequence #1 | This paper | CRISPR guide RNA sequence | GTCAGAGGAGCCAAAGCGCAGG |
| Sequence-based reagent | Mouse guide sequence #2 | This paper | CRISPR guide RNA sequence | GGCCCTCCTTCATCATCTGAA |
| Sequence-based reagent | Human PDEL HDR donor | This paper | CRISPR donor RNA sequence | GCCAAAGCCGGAAGAACCAAAGCGCAAG GTCGATAAGAAAAGAGCAAGGAGAGGGT CCTCTTCT |
| Sequence-based reagent | Mouse PDEL HDR donor | This paper | CRISPR donor RNA sequence | GAGGAGCAAAGGTCAGAGGAGCCAAAGC GCCGGGTGGATAAGAAGAGAGCGAGGAA GGCATCCTCTTCA |
| Sequence-based reagent | Mouse pMUT HDR donor | This paper | CRISPR donor RNA sequence | AGGTCAGAGGAGCCAAAGCGCAGGGTGG ATGCTGCAGTGGCTTGGGAGGCGTTGATTTGT GTCGGAGCGGCCAAGAAGAGAGCGAGGA AGGCATCCTCTTCA |
| Recombinant DNA reagent | OmicsLink expression clone of human AKAP12 in pRECEIVER-WG16 vector | Genecopoeia, MD | EX-H3212-WG16 | Vector for in vitro translation of AKAP12 controlled by T7 promoter |
| Recombinant DNA reagent | AAV-GFAP-Sacas9 | Vector Biolabs, PA | Cat #7125 | HSC-specific gene editing AAV vector |
| Recombinant DNA reagent | AAV-LRAT-Sacas9 | Vector Builder cloning service | | HSC-specific gene editing AAV vector |
| Peptide, recombinant protein | PKCα protein, active | MilliporeSigma, MA | 14-484 | In vitro kinase assay |
| Peptide, recombinant protein | HSP47 recombinant, human | Prospec NJ | HSP-047 | Recombinant binding assay |

*Continued on next page*

*Continued*

| Reagent type (species) or resource | Designation | Source or reference | Identifiers | Additional information |
|---|---|---|---|---|
| Chemical compound, drug | Carbon tetrachloride (CCl4) | Sigma-Aldrich | Cat #270652 | HPLC grade |
| Other | Lipofectamine RNAiMAX | Thermo Fisher Scientific | Cat #13778075 | Transfection of siRNA |
| Other | Dharmafect Duo reagent | Dharmacon, CO | Cat #T-2010-02 | Transfection of CRISPR components |
| Commercial assay or kit | QuikChange II site-directed mutagenesis Kit | Agilent Technologies, CA | Cat #200521 | Mutagenesis of AKAP12 plasmid |
| Commercial assay or kit | Non-radioactive TNT Coupled Transcription/Translation system | Promega, WI | Cat #L4610 | In vitro translation |
| Commercial assay or kit | Hydroxyproline Assay Kit | Cell Biolabs Inc, CA | Cat #STA-675 | Hydroxyproline measurement in liver |
| Commercial assay or kit | ALT colorimetric activity assay kit | Cayman Chemical, MA | Cat #700260 | ALT measurement in plasma |
| Commercial assay or kit | AST colorimetric activity assay kits | Cayman Chemical, MA | Cat #701640 | ALT measurement in plasma |
| Biological sample (*Homo sapiens*) | Primary human hepatic stellate cells | ScienCell Incorporation | Cat #5300 | |
| Biological sample (*H. sapiens*) | Human tissue array | Human tissue biorepository, US Biolabs Inc MD | XLiv086-01 | |
| Antibody | Anti-AKAP12 antibody (JP74 clone, mouse monoclonal) | Abcam | ab49849 | Immunoprecipitation: (1 µg/500 µg) extract; western: (1:2000) in 5% milk/TBS-Tween-20; PLA: (1:250) dilution in PLA buffer |
| Antibody | Anti-Phosphoserine antibody (rabbit polyclonal) | Abcam | ab9332 | PLA: (1:250) dilution |
| Antibody | Anti-α-SMA antibody (rabbit polyclonal) | Abcam | ab5694 | Western: (1:2000) in 5% milk/TBS-Tween-20 |
| Antibody | Anti-PKCα antibody (rabbit polyclonal) | Genetex | GTX130453 | Western: (1:2000) in 5% milk/TBS-Tween-20 |
| Antibody | Anti-Collagen I alpha (**Friedman, 2008**) antibody (COL-1 clone, mouse monoclonal) | Novus Biologicals | NB600-450 | Western: (1:1000) in 5% milk/TBS-Tween-20; PLA: (1:250) dilution in PLA buffer |
| Antibody | Anti-HSP47 antibody (clone # 950806, mouse monoclonal) | Novus Biologicals | MAB9166-100 | Western: (1:2000) in 5% milk/TBS-Tween; PLA: (1:250) dilution in PLA buffer |
| Antibody | Anti-Biotin antibody (rabbit polyclonal IgG) | Abcam | ab53494 | Western: (1:1000) in 5% milk/TBS-Tween-20 |
| Antibody | Anti-GAPDH antibody (rabbit polyclonal IgG) | Proteintech | 10494-1-AP | Western: (1:2000) in 5% milk/TBS-Tween |
| Antibody | Anti-SaCas9 antibody (Clone 11C12, mouse monoclonal) | Genetex | A01951 | Western: (1:2000) in 5% milk/TBS-Tween |
| Antibody | Anti-desmin antibody (rabbit polyclonal IgG) | Proteintech | 16520-1-AP | Immunostaining: (1:250) dilution in PLA buffer |
| Antibody | Anti-albumin antibody (rabbit polyclonal IgG) | Proteintech | 16475-1-AP | Immunostaining: (1:250) dilution in PLA buffer |
| Antibody | Anti-IRE1α antibody (rabbit polyclonal IgG) | Proteintech | 27528-1-AP | Western: (1:1000) in 5% milk/TBS-Tween-20; PLA: (1:250) dilution in PLA buffer |
| Antibody | Anti-Phospho-IRE1α (S724) antibody (rabbit polyclonal IgG) | Abcam | ab124945 | Western: (1:1000) in 5% BSA/TBS-Tween-20 |
| Antibody | Anti-phospho-Smad2 (Ser465/467)/Smad3 (Ser423/425) (rabbit polyclonal IgG) | Cell Signaling Technology | 8828 | Western: (1:2000) in 5% BSA/TBS-Tween-20 |
| Antibody | Anti-SMAD2 antibody (rabbit polyclonal IgG) | Proteintech | 12570-1-AP | Western: (1:2000) in 5% BSA/TBS-Tween-20 |
| Antibody | Anti-SMAD3 antibody (rabbit polyclonal IgG) | Proteintech | 25494-1-AP | Western: (1:2000) in 5% BSA/TBS-Tween-20 |
| Antibody | Phospho-p38 MAPK (Thr180/Tyr182) Antibody (rabbit polyclonal IgG) | Cell Signaling Technology | 9211 | Western: (1:2000) in 5% BSA/TBS-Tween-20 |
| Antibody | P38 MAPK Antibody (rabbit polyclonal IgG) | Cell Signaling Technology | 9212 | Western: (1:2000) in 5% BSA/TBS-Tween-20 |
| Antibody | Anti-BIP/GRP78 antibody (rabbit polyclonal IgG) | Proteintech | 11587-1-AP | Western: (1:2000) in 5% milk/TBS-Tween-20 |
| Antibody | Anti-IL1β antibody (rabbit polyclonal IgG) | Proteintech | 26048-1-AP | Western: (1:2000) in 5% milk/TBS-Tween-20 |
| Antibody | Anti-IL6 antibody (rabbit polyclonal IgG) | Proteintech | 21865-1-AP | Western: (1:2000) in 5% milk/TBS-Tween-20 |

*Continued*

| Reagent type (species) or resource | Designation | Source or reference | Identifiers | Additional information |
|---|---|---|---|---|
| Antibody | Anti-IL17 antibody (Clone 1B3D5, mouse monoclonal) | Proteintech | 66148-1-Ig | Western: (1:2000) in 5% milk/TBS-Tween-20 |
| Antibody | Anti-IL10 antibody (rabbit polyclonal IgG) | Proteintech | 20850-1-AP | Western: (1:2000) in 5% milk/TBS-Tween-20 |
| Antibody | Anti-calreticulin antibody (clone EPR3924, rabbit monoclonal) | Abcam | ab92516 | Immunostaining: (1:250) dilution in PLA buffer |
| Antibody | Clean-Blot IP Detection (HRP) (secondary antibody) | Life Technologies | 21230 | Detection: co-immunoprecipitation-immunoblot: (1:1000) in 5% milk/TBS-Tween-20 |
| Antibody | Streptavidin-HRP (secondary antibody) | Cell Signaling Technology | 3999 | Detection: Biotin western blots: (1:5000) in 5% milk/TBS-Tween-20 |
| Antibody | Goat anti rabbit IgG H&L (Alexa Fluor 488 green) (secondary antibody) | Abcam | ab150077 | Detection: immunoflorescence: (1:1000) in PLA buffer |
| Antibody | Goat Anti-Mouse IgG H&L (Alexa Fluor 488 green) (secondary antibody) | Abcam | ab150113 | Detection: immunoflorescence: (1:1000) in PLA buffer |
| Antibody | Goat Anti-Mouse IgG H&L (Alexa Fluor 647 far red) (secondary antibody) | Abcam | ab150115 | Detection: immunoflorescence: (1:1000) in PLA buffer |
| Antibody | Goat Anti-Rabbit IgG H&L (Alexa Fluor 647 far red) (secondary antibody) | Abcam | ab150079 | Detection: immunoflorescence: (1:1000) in PLA buffer |
| Antibody | Duolink In Situ PLA Probe Anti-Mouse PLUS (secondary antibody) | MilliporeSigma | DUO92001 | Detection: PLA: (1:600) in PLA buffer |
| Antibody | Duolink In Situ PLA Probe Anti-Mouse MINUS (secondary antibody) | MilliporeSigma | DUO92004 | Detection: PLA: (1:600) in PLA buffer |

## Primary cell isolation and culture

Primary human HSCs purchased from ScienCell Incorporation (CA) were cultured on plastic dishes for 6 hr (Day 0) or further cultured till activation (Days 5–7). Mouse HSCs or hepatocytes were isolated from 3 to 4 months old C57BL/6 mice according to our previously established protocols (*Ramani et al., 2018*). Mouse HSCs were culture-activated on plastic dishes like human HSCs.

## Phospho-peptide mapping

AKAP12 was immunoprecipitated from HSCs or hepatocytes using an AKAP12 antibody-conjugated protein A/G column (Thermo Fisher Scientific). The AKAP12 beads were submitted to Applied Biomics, CA for phospho-peptide mapping. Tryptic peptides were enriched for phospho-peptides and processed for detection of a phospho-site by mass spectrometry. Phosphorylated residues were confirmed by mass spectrometry peak showing the neutral loss of phosphate that was detected from peak shifts on MS/MS spectrum (*Table 1—source data 1*). The observed mass of a phospho-peptide was reduced by 98 Da if a single serine/threonine showed a neutral loss of phosphate.

## CRISPR gene editing in cultured HSCs

CRISPR-Cas9 mediated gene editing at the AKAP12 gene locus (exon 3) to delete the region of its activation-responsive phospho-sites was performed by HDR. A 22-bp small guide RNA sequence (sgRNA) upstream of a protospacer adjacent motif (PAM- 5′-GTGGAT-3′) recognized by saCas9 (PAM consensus-NNGRRT where N=any nucleotide, R=A or G) (*Xie et al., 2018*), was designed and synthesized using the Edit-R CRISPR system (Horizon Discovery, CO) (human guide sequence, Key resource table). The CRISPR design tool was used to determine the sgRNA whose sequence is unique compared to the rest of the genome to avoid off-target effects. A donor RNA to delete the phospho-region was designed and synthesized using the Edit-R HDR donor designer system (Horizon) (human PDEL HDR donor, Key resource table). The sgRNA was stabilized by 2′-O-methyl nucleotides and phosphorothioate linkages in the backbone on both the 5′ and 3′ ends and the HDR donor was stabilized by phosphorothioate linkages on both ends to improve functionality during transfection. Cultured cells were co-transfected with a commercially available plasmid, AAV6-GFAP-saCas9, containing the SaCas9 gene under control of the GFAP promoter (Vector Biolabs, PA), sgRNA and HDR donor RNA using the DharmaFECT Duo Transfection Reagent that allows co-transfection of RNA and DNA (Horizon). Cells with transfection reagent alone or SaCas9 plasmid alone +transfection reagent were used as controls. CRISPR designs for mouse HSCs were performed as above for human with mouse guide sequence #1

and mouse PDEL HDR donor (Key resource table). After 48–72 hr of transfection, genomic DNA from human or mouse HSCs was amplified by multiplex PCR using two primers to amplify the region around the deletion site and a third deletion-specific primer to detect HDR-mediated gene editing.

## Gene silencing in activated HSCs

Activated human HSCs (0.3 million cells per well of six-well plate) were reverse transfected with a universal negative control (Cat #4404021), *Prkca* (Cat #s11092), or *Prkca* B (Cat #s11094) silencerselect siRNA (Thermo Fisher Scientific, IL) using the lipofectamine RNAiMAX reagent as we described previously (*Ramani et al., 2018*).

## Carbon-tetrachloride (CCl₄) injection in mice

About 12-week-old C57BL/6 male mice were injected intraperitoneally with CCl$_4$ (HPLC grade, Cat #270652, Sigma-Aldrich, diluted 1:3 in mineral oil) or mineral oil (control) at 1 µl/gram body weight bi-weekly for 5 weeks. All procedures for the care and use of mice were approved by the Institutional Animal Care and Use Committee at Cedars-Sinai Medical Center (CSMC).

## CRISPR gene editing in mice

HDR-based gene editing in control or CCl$_4$ mice was performed according to the scheme in *Figure 4A and B*. Two 22-bp sgRNA sequences upstream of a saCas9 PAM (*Xie et al., 2018*), were designed using the Edit-R CRISPR system (Horizon Discovery). Off-target analysis for the two sgRNA was performed using the algorithm from the Benchling (Biology Software-(2022) retrieved from https:// benchling.com) (*Supplementary file 5*). The two sgRNA sequences were cloned into a single AAV6 vector under the control of a U6 promoter by the cloning service available from Vector builder Inc, IL. An AAV6 vector containing a non-targeting sgRNA was used as an EV control. The sequence corresponding to a PDEL or PMUT donor with 500 bp flanking either side of the target region was cloned into a separate AAV6 vector. The PAM sequence in these donors was mutated to prevent re-cleavage by SaCas9 after HDR. The AAV6-GFAP-SaCas9 vector (Vector Biolabs) was used for HSC-specific gene editing. In addition, another AAV6-LRAT-SaCas9 vector was prepared by cloning the mouse LRAT promoter (Accession ID: NM_023624) upstream of SaCas9 (Vector Builder). AAV6 particles of the sgRNA construct, EV construct, PDEL/PMUT donors, and GFAP/LRAT-SaCas9 were purified using Vector builder's AAV production service. For each viral vector, titer was determined by real-time PCR using primers specific for the AAV inverted terminal repeats (ITRs). A titer of 1–2×10$^{13}$ genome copies (GCs)/ml was achieved for each AAV. All vectors tested negative for mycoplasma contamination. EV or sgRNA vectors along with PDEL or PMUT donors and GFAP or LRAT SaCas9, were injected into tail vein of mice at 10$^{11}$ GC/vector in a volume of 100 µl phosphate-buffered saline. Viral vectors were injected into oil or CCl$_4$ mice during the second and fourth week of oil or CCl$_4$ administration (*Figure 4B*). The HSC specificity of CRISPR was determined by SaCas9 immunofluorescence as described under the Immunostaining section. The efficiency of CRISPR editing in HSCs and hepatocytes of gene-edited livers was evaluated by NGS. A 298-bp PCR product was amplified from genomic DNA using primers that recognized regions upstream and downstream of the site of AKAP12 deletion or mutation. Amplicons were purified from gels and submitted to Azenta Life Sciences Inc, CA. for performing NGS. Briefly, Illumina adaptor sequences (FW: 5'-ACACTCTTTCCCTACACGACGCTC TTCCGATCT-3', REV: 5'-GACTGGAGTTCAGACGTGTGCTCTTCCGA TCT-3') were added to the amplicons and sequenced by Azenta Illumina platform sequencers. The WT and mutant or deletion mutant reads were counted from each sample and the efficiency of editing was the percentage of edited reads (PDEL or PMUT) versus the total reads. Frequencies of on-target and off-target base changes were analyzed by comparing the target reads to reference reads corresponding to the WT Akap12 amplicon between the two sgRNA sequences (*Figure 4A*). Within this region, any mismatches other than PDEL or PMUT were considered as off-targets. The mismatches to the reference were observed mainly outside the target region at a frequency of 5% or less (*Figure 4G*).

## Human tissue array

The human tissue array (Cat #XLiv086-01) in the form of paraffin-embedded tissues was purchased from the human tissue biorepository, US Biolabs Inc, MD. Arrays were stained by immunostaining as described below.

## Real-time RT-PCR

Total RNA from cells or tissues was reverse transcribed to cDNA using M-MLV reverse transcriptase (Nxgen). CDNA was subjected to quantitative RT-PCR using TaqMan probes for mouse *Akap12*, *Col1a1*, *Acta2*, and the housekeeping gene, *Gapdh* (mouse) (Life Technologies) (*Ramani et al., 2018*). The PCR profile was: initial denaturation: 95°C for 3 min, 45 cycles: 95°C, 3 s; 60°C, 30 s. The cycle threshold (Ct value) of the target genes was normalized to that of control gene to obtain the delta Ct ($\Delta$Ct). The $\Delta$Ct was used to find the relative expression of target genes according to the formula: relative expression=$2^{-\Delta\Delta Ct}$, where $\Delta\Delta$Ct=$\Delta$Ct of target genes in experimental condition − $\Delta$Ct of target gene under control condition.

## Co-immunoprecipitation and western blotting

Total protein extract was processed for immunoprecipitation by incubating 200 µg of pre-cleared protein with 2 µg of antibody as we described previously (*Ramani et al., 2018*). Immunoprecipitated protein was processed for western blotting as previously published (*Ramani et al., 2018*) and developed with Clean-blot IP detection reagent (HRP) (Thermo Fisher Scientific, IL). Antibodies used for western blotting are listed in Key resource table.

## Vitamin A autofluorescence

UV-excited autofluorescence of human HSCs was captured by fluorescence microscopy using a Keyence BZ-X710 inverted fluorescent microscope (Itasca, IL) as we described previously (*Ramani et al., 2018*).

## Site-directed mutagenesis

An expression vector (pReceiver-WG16) containing the human *AKAP12* gene under control of the T7 promoter was purchased from Genecopoiea, MD and mutated at AKAP12's activation-responsive sites (S/T to A mutations) using the QuikChange II site-directed mutagenesis kit (Agilent Technologies, CA) as we described previously (*Ramani et al., 2015*). Mutations were detected by sequencing the clones at the Azenta DNA sequencing facility using an *AKAP12* gene-specific primer (5′-GAGAAGGT GTCACTCCC-3′).

## In vitro kinase assay, phostag analysis, and binding studies

The T7-AKAP12 vector or its mutants were in vitro translated using the non-radioactive TNT Coupled Transcription/Translation system containing rabbit reticulocyte lysate (RRL) and a biotin-lysyl tRNA according to the manufacturer's instructions (Promega, WI) to incorporate biotin label into the translated AKAP12 protein. Biotinylated AKAP12 was purified from the RRL components using a biotin-antibody column. Biotinylated AKAP12 or its mutants (5 µl) were used as a substrate for PKCα in a 25-µl in vitro kinase reaction using 100 ng of active recombinant PKCα enzyme (MilliporeSigma, MA), 5 µl of a lipid activator (MilliporeSigma; 20 mM MOPS, pH 7.2, 25 mM β-glycerolphosphate, 1 mM sodium orthovanadate, 1 mM dithiothreitol, and 1 mM CaCl2), 3 µl of $Mg^{2+}$/ATP cocktail (MilliporeSigma, 20 mM MOPS, pH 7.2, 25 mM β-glycerophosphate, 5 mM EGTA, 1 mM $Na_3VO_4$, 1 mM dithiothreitol, 75 mM $MgCl_2$, and 0.5 mM ATP) and 2.5 µl of 20 mM Hepes-NaOH buffer, pH 7.6. The reaction was carried out at 30°C for 2 hr. The kinase reaction was run on a zinc phostag gel containing 15 µM phostag gel (Fujifilm Wako Chemicals, VA) to separate phosphorylated form of AKAP12 from its unphosphorylated counterparts as we described earlier (*Ramani et al., 2015*). Membranes were probed with streptavidin-HRP (Key resource table) to detect biotinylated AKAP12. Biotin antibody was conjugated to protein A/G plus agarose columns using a coupling buffer according to the crosslink immunoprecipitation kit (Thermo Fisher Scientific) followed by binding of recombinant biotinylated AKAP12. The columns were treated with recombinant HSP47 protein in the absence or presence of active PKCα enzyme. Bound proteins were eluted from the washed column using elution buffer from the crosslinking immunoprecipitation kit (Thermo Fisher Scientific) and run on gels along with biotinylated AKAP12 as input and antibody-bound protein A/G beads as IgG controls. Blots were incubated with HSP47 antibody followed by Clean-blot IP detection. Reverse IP was done by following the same protocol using HSP47 antibody columns treated with biotinylated AKAP12 followed by detection with streptavidin-HRP. Recombinant HSP47 input was purchased from Prospec protein specialists, NJ.

## Duolink PLA and immunostaining procedures

For immunocytochemical procedures, cells were fixed with paraformaldehyde and then permeabilized with Triton-X 100 before antibody staining. For immunohistochemical analysis, tissues were de-par-affinized and antigen retrieval was performed using the citrate-based antigen unmasking solution (Vector Laboratories, CA). For phospho-detection using PLA, primary AKAP12 or phospho-serine (PSer) antibodies (see Key resource table) were directly conjugated to PLA minus or plus complementary oligonucleotide arms (PLA minus, Catalog no. DUO92010; PLA plus, Catalog no. DUO92009, MilliporeSigma) according to our previously published protocol (*Ramani et al., 2018*). To examine protein-protein interactions in cells or tissues, samples were incubated with the antibodies for the interacting targets at 4°C overnight (AKAP12-HSP47, HSP47-collagen). After washing the unbound antibodies, samples were further incubated overnight with secondary antibodies (rabbit or mouse) that were bound to PLA plus or minus complementary probes (MilliporeSigma, Key resource table). The PLA probes were ligated when the proteins were in proximity due to their interaction giving a fluorescent signal as we previously reported (*Ramani et al., 2018*). To evaluate the localization of interacting partners, co-immunostaining of the PLA signals was done with HSC (desmin) or subcellular compartment (calreticulin ER) marker antibodies. Marker antibodies were detected by Alexa fluor green rabbit or mouse secondary antibodies (Abcam, Key resource table). Co-localization of SaCas9 with desmin or albumin markers in liver tissue was detected by Alexa fluor secondary antibodies (see Key resource table). AKAP12 expression in tissues was detected using the mouse HRP/DAB detection immunohistochemistry kit (Cat #ab64264, Abcam).

## Histopathological examination

Liver sections fixed with 10% neutral formalin were processed for paraffin embedding, sectioning, H&E, and picrosirius red staining (collagen) using the services provided by the liver histology core of the University of Southern California research center for liver diseases (NIH grant P30 DK048522).

## Hydroxyproline measurement

The hydroxyproline content of tissue was measured following the protocol from the hydroxyproline assay kit (Cell Biolabs Inc, CA). Briefly, 10 mg of liver tissue was homogenized, and acid hydrolysis was done with 12 N HCl. Hydrolyzed samples were treated with chloramine T to convert the hydroxypro-line to a pyrrole. Ehrlich's reagent or 4-(Dimethylamino) benzaldehyde added to the pyrrole reacted with it to produce a chromophore whose absorbance could be read at 540–560 nm. The content of hydroxyproline in the tissue sample was determined by comparison to a hydroxyproline standard from the kit that was processed like the unknown sample.

## ALT/AST measurement

ALT and AST levels from plasma of mice were measured with the ALT and AST colorimetric activity assay kits (Cayman Chemical, MI). ALT activity was measured by monitoring the rate of NADH oxida-tion in a coupled reaction using lactate dehydrogenase (LDH). The NADH to NAD+ oxidation caused a decrease in A340 nm absorbance. The rate of decrease (ΔA340/min) is directly proportional to the ALT activity. AST activity was measured by the rate of NADH oxidation in the presence of malate dehy-drogenase. NADH to NAD+ conversion caused a decrease in A340 nm absorbance. LDH was added to the AST reaction to prevent interference from endogenous pyruvate in the plasma. The ΔA340/min for both ALT and AST were converted to units/L by dividing the ΔA340 values by the NADH extinction coefficient and multiplying by the sample dilution factor as per the protocol instructions (Cayman).

## Proteomics analysis

Total protein from liver or HSCs was subjected to mass spectrometry-based proteomics analysis by the services of Poochon proteomics solutions, MD. The Nanospray LC/MS/MS analysis of tryptic peptides for each sample was performed sequentially with a blank run between each two sample runs using a Thermo Scientific Orbitrap Exploris 240 Mass Spectrometer and a Thermo Dionex Ulti-Mate 3000 RSLCnano System. Peptides from trypsin digestion were loaded onto a peptide trap cartridge at a flow rate of 5 μl/min. The trapped peptides were eluted onto a reversed-phase Easy-Spray Column PepMap RSLC, C18, 2 μM, 100 A, 75 μm×250 mm (Thermo Fisher Scientific, CA) using a linear gradient of acetonitrile (3–36%) in 0.1% formic acid. The elution duration was 110 min

at a flow rate of 0.3 µl/min. Eluted peptides from the Easy-Spray column were ionized and sprayed into the mass spectrometer, using a Nano Easy-Spray Ion Source (Thermo Fisher Scientific) under the following settings: spray voltage, 1.6 kV, Capillary temperature, 275°C. Other settings were empirically determined. Raw data files were searched against mouse protein sequences database using the Proteome Discoverer 1.4 software (Thermo Fisher Scientific) based on the SEQUEST algorithm. Carbamidomethylation (+57.021 Da) of cysteines was set as fixed modification, and Oxidation/+15.995 Da (M), and Deamidated/+0.984 Da (N, Q) were set as dynamic modifications. The minimum peptide length was specified to be five amino acids. The precursor mass tolerance was set to 15 ppm, whereas fragment mass tolerance was set to 0.05 Da. The maximum false peptide discovery rate was specified as 0.05. The resulting Proteome Discoverer Report contains all assembled proteins with peptides sequences and peptide spectrum match counts (PSM#). The PSM count is a measure of the abundance of the protein.

## Statistical analysis

Western blotting data were quantified by densitometry of blots using the ImageJ software (NIH). PLA staining and immunofluorescence data were analyzed in a blinded manner by two individuals and quantified using ImageJ according to published protocols (*López-Cano et al., 2019*). Scatter bars showing individual experimental points and their means were plotted using GraphPad Prism 9.3.0, GraphPad software. Biologically independent replicates combined from at least three individual experiments were represented as mean ± standard error (mean ± SE). Statistical analysis was performed using two-tailed Student's t-test for paired comparisons and one-way ANOVA (GraphPad Prism) for comparing differences between multiple groups. Significance was defined as $p < 0.05$. Tukey HSD post hoc test for each comparison is shown as source data.

## Acknowledgements

This work was supported by NIH grants 1R21ES030534-01A1 (K Ramani) and 1 R21AA027352-01A1 (ML Tomasi, K Ramani).

## Additional information

### Funding

| Funder | Grant reference number | Author |
|---|---|---|
| National Institutes of Health | 1R21ES030534-01A1 | Komal Ramani |
| National Institutes of Health | 1 R21AA027352-01A1 | Maria Lauda Tomasi |

The funders had no role in study design, data collection and interpretation, or the decision to submit the work for publication.

### Author contributions

Komal Ramani, Conceptualization, Resources, Funding acquisition, Validation, Investigation, Methodology, Writing - original draft; Nirmala Mavila, Formal analysis, Methodology, Writing – review and editing; Aushinie Abeynayake, Validation, Investigation, Methodology; Maria Lauda Tomasi, Formal analysis, Writing – review and editing; Jiaohong Wang, Michitaka Matsuda, Investigation, Methodology; Eki Seki, Resources, Formal analysis

### Author ORCIDs

Komal Ramani http://orcid.org/0000-0002-2387-4603
Maria Lauda Tomasi http://orcid.org/0000-0001-8156-9052

### Ethics

This study was performed in strict accordance with the recommendations in the Guide for the Care and Use of Laboratory Animals of the National Institutes of Health. All procedures for the care and use

of mice were approved by the Institutional Animal Care and Use Committee at Cedars-Sinai Medical Center (CSMC) under protocol # IACUC008834.

### Decision letter and Author response

Decision letter https://doi.org/10.7554/eLife.78430.sa1
Author response https://doi.org/10.7554/eLife.78430.sa2

## Additional files

### Supplementary files

• Supplementary file 1. Kinase-prediction for AKAP12's activation-responsive phospho-sites.

• Supplementary file 2. Datasets of next-generation amplicon sequencing (NGS) from PDEL CRISPR mouse model. Genomic DNA of HSCs isolated from oil or $CCl_4$ injected mice treated with AKAP12 PDEL CRISPR +GFAP-Cas9 or LRAT-Cas9 were submitted for NGS to Azenta Life Sciences as described under methods. Hepatocytes from PDEL CRISPR +GFAP-Cas9 were also processed as above for NGS. Representative raw reads of WT, deletion or base changes are shown for each data set and summarized in the first summary tab of the excel.

• Supplementary file 3. Datasets of next-generation amplicon sequencing (NGS) from PMUT CRISPR mouse model. Genomic DNA of HSCs or hepatocytes isolated from oil or $CCl_4$ injected mice treated with AKAP12 PMUT CRISPR +GFAP-Cas9 were submitted for NGS to Azenta Life Sciences as described under methods. Representative raw reads of WT or base changes are shown for each data set and summarized in the first summary tab of the excel.

• Supplementary file 4. Proteomics analysis of total liver and HSCs from CRISPR PDEL mouse model. Total protein from the liver or HSCs of AKAP12 PDEL CRISPR +GFAP-Cas9 mice was subjected to mass spectrometry-based proteomics analysis as described under methods. Proteomics dataset of whole liver, ER stress/UPR components of the liver and HSCs is shown. The summary tab in the excel explains each dataset.

• Supplementary file 5. Mouse sgRNA off-target analysis.

• Transparent reporting form

### Data availability

All data generated or analysed during this study are included in the manuscript and supporting file; Source Data files have been provided for Figures 1, 2, 3, 4, 5,6, 7.

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
