## [Editor Report]

Liver fibrosis is a complication of diverse liver disorders, including fatty liver disease, and in this important work, Ramani et al. provide solid evidence that AKAP12 enhances collagen production in a CCL4-induced mouse model of liver fibrosis, and they demonstrate that in hepatic stellate cells (HSCs) AKAP12 is phosphorylated by PKCalpha, which, in turn, leads to increased scaffolding activity towards HSP47, a chaperone of collagen located in the endoplasmic reticulum (ER). AKAP12 activation also resulted in increased ER stress and the generation of inflammatory mediators, and targeting AKAP12 phosphorylation sites in HSCs resulted in suppression of collagen synthesis and ER stress and fibrotic response in the mice. The important findings of the study include the identification of a novel disease mechanism and an attractive drug target for liver fibrosis. The finding should, however, be cautiously interpreted as the role of AKAP12 in liver fibrosis has to be explored in other models of liver fibrosis.

---

## [Decision Letter]

**Decision letter after peer review:**

Thank you for submitting your article "Targeting A-Kinase Anchoring Protein 12 Phosphorylation in Hepatic Stellate Cells Regulates Liver Injury and Fibrosis in Mouse Models" for consideration by *eLife*. Your article has been reviewed by 3 peer reviewers, and the evaluation has been overseen by a Reviewing Editor and Jonathan Cooper as the Senior Editor. The following individual involved in the review of your submission has agreed to reveal their identity: Johannes Broichhagen (Reviewer #1).

Revisions should include the following, in addition to a point-by-point response to the comments made by the 3 reviewers.

– Several protein blots, as outlined by reviewer 2, should be redone. Specifically replace Figure 1 A and B. Please specify if the difference in Akap12 expression shown in Figure 1A is significant.

– The number of mice for comparing CR-PMUT to CR-PDEL should be increased to at least 6 for both groups.

– Add scale bars to all immunofluorescence images.

– Figure 5B: Please show Sirius red as well as HandE.

– Transcript levels for collagen and a-SMA should be provided.

– Please provide a time course of AKAP12-HSP47 interaction during in vitro activation of HSC.

– Please follow our advice about statistical reporting.

*Reviewer #1 (Recommendations for the authors):*

As mentioned in the public statement, I only feel confident in recommending improvements from a technical side. Some issues need to be addressed in detail from my point of view.

– Revise Figure 7F to contain only information obtained in this study.

– Some blots seem saturated, which should be double-checked by authors. Especially since they perform quantitative analyses.

– Scale bars are missing in the microscopic images and should be added.

– I find it difficult to interpret significance descriptors, i.e. which data is compared to which in the plots. A clearer annotation using horizontal bars could be helpful. In a similar vein, is the difference in Akap12 expression significant in Figure 1A? Otherwise, it would help to label non-significant differences with "n.s." or the like.

– The many tables in Figure 4G and in Figure 7 could be moved to the Supporting Information to help balance the manuscript.

*Reviewer #2 (Recommendations for the authors):*

Although the study is interesting and reports a novel mechanism for drug design to potentially target the process of liver fibrosis, several issues related to data presentation call for questions on the rigor of the study:

1. Figure 1A. Blots in figure 1A for AKAP12 and GAPDH are either modified (bands appear "squished" on the vertical axis in figure 1A when compared to the source data). Moreover, the appearance of the AKAP12 bands on the source data figure would suggest that the protein lysate was not separated for the appropriate amount of time as the bands look to not have entered the separating portion of the gel. In other words, the proteins appear to be "stuck" in the stacking part of the acrylamide gel. This western blot should be redone.

2.Figure 1B. Immunofluorescence is not of good quality (over-exposed) and merged panels are not the same exact field as the single staining. Also, contrary to what is expected from a PLA staining, i.e small fluorescent dots that denote interaction between two antigens/proteins of interest, figures show large "blobs" as positive interaction. Based on this, the conclusion that the images display AKAP12 phosphorylation does not seem accurate. Also, 200x merged figure is not aligned with the three channels displayed. A similar issue is present in additional images provided as source data. Is 400x from the same field? If so, please indicate the area on the merged figure from the 200x magnification that was used to produce the 400x images. Similarly, for AKAP12/HRP immunohistochemistry 400x and 200x are from different fields.

3. It would be interesting to show by WB how the AKAP12-HSP47 interaction and the aSMA levels change during in vitro activation of HSC (from day0 to day5) and perhaps add to figure 3.

4. In figure 4, a better estimation of the in vivo efficiency of AAV6 serotype vectors in PDL or PMUT donors should be performed by quantifying the cell numbers per field instead of the overall fluorescence signal count.

5. Source data figure 5B. Some HandE pictures that are displayed as belonging to different mice are clearly nearby fields of the same mouse liver. Please change as according to the figure caption.

6. Figure 6D. OIL-OCR-PDEL and OIL-CR-PMUT appear to show the same picture with different magnifications and not different treatments as stated.

7. Figure 6E. Sirius red images from OIL-EV mice (I-II-III) and from OIL-CR-PMUT (II-III) have the same issue as in point 5. Pictures that are displayed as representing different mice appear to be slightly different fields of the same mouse.

8. The decreased effect of CR-PMUT compared to CR-PDEL is possibly due to lower statistical power in the former (n=3 in CR-PMUT compared to n=6 in CR-PDEL). The author should increase the number of mice.

9. The role of PKC α should be better contextualized and expanded in the discussion.

10. Scale bars should be added to all immunofluorescence images.

11. Post-test used should be indicated for the ANOVA statistic.

---

## [Author Response]

Revisions should include the following, in addition to a point-by-point response to the comments made by the 3 reviewers.– Several protein blots, as outlined by reviewer 2, should be redone. Specifically replace Figure 1 A and B. Please specify if the difference in Akap12 expression shown in Figure 1A is significant.

The western blots have been repeated and data is now presented in revised figure 1A. The statistical significance is indicated. Revised source data is presented in figure 1-source data 1. Figure 1B is also repeated as per recommendation from reviewer 2.

– The number of mice for comparing CR-PMUT to CR-PDEL should be increased to at least 6 for both groups.

The number of mice per group has been increased to 6.

– Add scale bars to all immunofluorescence images.

Scale bars are added to all immunofluorescence images.

– Figure 5B: Please show Sirius red as well as HandE.

We had originally shown Sirius red staining in figure 6 but have now included it along with HandE in a revised figure 5D in the revised manuscript. Source data for the same is provided.

– Transcript levels for collagen and a-SMA should be provided.

Transcript levels for *Col1A1* (collagen mRNA) and *Acta2* (α-sma mRNA) have been included in revised Figure 6D of the manuscript.

– Please provide a time course of AKAP12-HSP47 interaction during in vitro activation of HSC.

Time course experiments for AKAP12-HSP47 interaction are now shown in revised figure 2A of the manuscript.

– Please follow our advice about statistical reporting.

We have now included post-hoc analysis for each data in the manuscript. The post-hoc analysis is uploaded as source data for each figure. Statistical indicators are clearly shown on each graph for comparison. In some cases, where space is limited, a statistical legend is provided below the graph.

A key resource table has been included in the main manuscript before the Materials and methods section.

Reviewer #1 (Recommendations for the authors):As mentioned in the public statement, I only feel confident in recommending improvements from a technical side. Some issues need to be addressed in detail from my point of view.– Revise Figure 7F to contain only information obtained in this study.

Following the reviewer’s advice, we have now modified figure 7F simply as a summary of our findings instead of a proposed model.

– Some blots seem saturated, which should be double-checked by authors. Especially since they perform quantitative analyses.– Scale bars are missing in the microscopic images and should be added.

Scale bars have now been added to the immunofluorescence images.

– I find it difficult to interpret significance descriptors, i.e. which data is compared to which in the plots. A clearer annotation using horizontal bars could be helpful. In a similar vein, is the difference in Akap12 expression significant in Figure 1A? Otherwise, it would help to label non-significant differences with "n.s." or the like.

Based on the concern expressed by the reviewer, statistical indicators are now clearly shown on the revised graphs for comparison. In some cases, where space is limited, a statistical legend is provided below the graph. For insignificant changes, “NS” is used. In addition, we have now included post-hoc analysis for each data in the manuscript. The post-hoc analysis is uploaded as source data for each figure. Figure 1A (mRNA) level is insignificant between oil and CCl4 and have now marked this as NS.

– The many tables in Figure 4G and in Figure 7 could be moved to the Supporting Information to help balance the manuscript.

We have retained the table in figure 4G as a summary of the NGS data since it does not take up much space. However, following the reviewer’s advice, we have removed tables from revised figure 7 to help balance the figure. Instead, we have the raw data of each experiment represented as a graph in revised figure 7—figure supplement 1. In addition, we have post-hoc analysis source data for each of the sub-figures of figure 7.

Reviewer #2 (Recommendations for the authors):Although the study is interesting and reports a novel mechanism for drug design to potentially target the process of liver fibrosis, several issues related to data presentation call for questions on the rigor of the study:1. Figure 1A. Blots in figure 1A for AKAP12 and GAPDH are either modified (bands appear "squished" on the vertical axis in figure 1A when compared to the source data). Moreover, the appearance of the AKAP12 bands on the source data figure would suggest that the protein lysate was not separated for the appropriate amount of time as the bands look to not have entered the separating portion of the gel. In other words, the proteins appear to be "stuck" in the stacking part of the acrylamide gel. This western blot should be redone.

The western blots have been repeated and data is now presented in revised figure 1A. The statistical significance is indicated. Revised source data is presented in figure 1-source data 1.

2.Figure 1B. Immunofluorescence is not of good quality (over-exposed) and merged panels are not the same exact field as the single staining. Also, contrary to what is expected from a PLA staining, i.e small fluorescent dots that denote interaction between two antigens/proteins of interest, figures show large "blobs" as positive interaction. Based on this, the conclusion that the images display AKAP12 phosphorylation does not seem accurate. Also, 200x merged figure is not aligned with the three channels displayed. A similar issue is present in additional images provided as source data. Is 400x from the same field? If so, please indicate the area on the merged figure from the 200x magnification that was used to produce the 400x images. Similarly, for AKAP12/HRP immunohistochemistry 400x and 200x are from different fields.

We thank the reviewer for pointing out these errors. Usually, PLA staining for interaction gives small dots in cells and a mixture of small and medium dots when staining tissue. In this case we performed a PLA of AKAP12 with phospho-serine antibody to identify phospho-serine phosphorylation of AKAP12 in tissues. This antibody combination with phospho-serine antibody stained as large blobs of positive interaction in the tissues we examined. To address this concern of the reviewer, we have now repeated the phospho-serine/AKAP12 PLA with OIL and CCL4 livers that were recently prepared from new experiments. The PLA staining is better than the old experiments and so we have included it in revised figure 1B and included source data for the same.

In addition, we now have the three channels aligned for figure 1B and source data.

For the IHC staining, we have now shown 200X and 400X from the same field and marked the area of 200X that was magnified to show 400X.

We hope that this alleviates the above concerns.

3. It would be interesting to show by WB how the AKAP12-HSP47 interaction and the aSMA levels change during in vitro activation of HSC (from day0 to day5) and perhaps add to figure 3.

We thank the reviewer for this suggestion and have now included data on AKAP12-HSP47 interaction in an HSC time course of day 0 to day 6. Data is presented in revised figure 2A.

4. In figure 4, a better estimation of the in vivo efficiency of AAV6 serotype vectors in PDL or PMUT donors should be performed by quantifying the cell numbers per field instead of the overall fluorescence signal count.

We thank the reviewer for this suggestion and have now estimated the PDEL or PMUT AAV efficiency by quantifying the SaCas9-positivity per desmin field. Data is now presented in revised figures 4D and 4F.

5. Source data figure 5B. Some HandE pictures that are displayed as belonging to different mice are clearly nearby fields of the same mouse liver. Please change as according to the figure caption.

We are thankful to the reviewer for checking this out. Yes, some pictures are nearby areas of the same liver as opposed to a different mouse liver. We have now fixed this problem by re-checking all the raw images. We have now included these in revised figure 5-source data 1.

6. Figure 6D. OIL-OCR-PDEL and OIL-CR-PMUT appear to show the same picture with different magnifications and not different treatments as stated.

Thanks for pointing this out. We agree that they look similar, so we have checked the image labels to figure out the error. We have revised the image and presented as revised figure 6E (originally figure 6D) in the manuscript.

7. Figure 6E. Sirius red images from OIL-EV mice (I-II-III) and from OIL-CR-PMUT (II-III) have the same issue as in point 5. Pictures that are displayed as representing different mice appear to be slightly different fields of the same mouse.

We have gone back to our original images to check this and have fixed this issue by showing the correct mouse liver for OIL+EV and OIL+CR-PMUT. Following recommendation from reviewer 3, we have shifted the Sirius red images along with HandE to figure 5, so this change will be reflected in the revised figure 5D.

8. The decreased effect of CR-PMUT compared to CR-PDEL is possibly due to lower statistical power in the former (n=3 in CR-PMUT compared to n=6 in CR-PDEL). The author should increase the number of mice.

We agree with the reviewer and have performed more CR-PMUT experiments and added them to the revised manuscript. Our overall analysis on the effect of CR-PMUT shows that some parameters such as AST levels are substantially suppressed very similar to PDEL; however, other parameters such as hydroxyproline (collagen quantification) level are reduced but not completely normalized like PDEL. All statistical correlations between the different groups are now clearly shown in the graphs of PDEL and PMUT and post-hoc analysis has been done and presented as source data for each figure.

9. The role of PKC α should be better contextualized and expanded in the discussion.

We thank the reviewer for this suggestion and have revised the discussion explaining the role of PKC-α and how it can be useful for future drug design targeting AKAP12 phosphorylation.

10. Scale bars should be added to all immunofluorescence images.

Scale bars are now included in the revised images

11. Post-test used should be indicated for the ANOVA statistic.

Post-hoc tests are now included as source data.